# α-Synuclein fibrils enhance HIV-1 infection of human T cells, macrophages and microglia

Lia-Raluca Olari [1,7], Sichen Liu [1,7], Franziska Arnold[1], Julia Kühlwein[2], Marta Gil Miró[1], Ajeet Rijal Updahaya[3], Christina Stürzel[1], Dietmar Rudolf Thal [3,4], Paul Walther[5], Konstantin M. J. Sparrer [1,6], Karin M. Danzer [2,6], Jan Münch [1] & Frank Kirchhoff [1] ✉

HIV-associated neurocognitive disorders (HAND) and viral reservoirs in the brain remain a significant challenge. Despite their importance, the mechanisms allowing HIV-1 entry and replication in the central nervous system (CNS) are poorly understood. Here, we show that α-synuclein and (to a lesser extent) Aβ fibrils associated with neurological diseases enhance HIV-1 entry and replication in human T cells, macrophages, and microglia. Additionally, an HIV-1 Env-derived amyloidogenic peptide accelerated amyloid formation by α-synuclein and Aβ peptides. Mechanistic studies show that α-synuclein and Aβ fibrils interact with HIV-1 particles and promote virion attachment and fusion with target cells. Despite an overall negative surface charge, these fibrils facilitate interactions between viral and cellular membranes. The enhancing effects of human brain extracts on HIV-1 infection correlated with their binding to Thioflavin T, a dye commonly used to stain amyloids. Our results suggest a detrimental interplay between HIV-1 and brain amyloids that may contribute to the development of neurodegenerative diseases.

Amyloid fibrils are best known for their role in neurodegenerative disorders, such as Alzheimer's, Parkinson's, and Huntington's disease[1–4]. It has become clear, however, that amyloids exert important physiological functions[5,6] and are associated with numerous human diseases[7–9]. For example, amyloid aggregates might promote HIV-associated neurological disorders (HAND) that are observed in about 30–50% of infected individuals and frequently persist even under effective combined antiretroviral therapy (cART)[10–13]. Neurocognitive impairment is observed in people living with HIV (PLWH) at a median age of 46, much earlier than in uninfected individuals. The most severe form of HAND called HIV-associated dementia (HAD), is associated with cognitive and motor dysfunctions similar to dementia related to Alzheimer's disease[14,15]. Deposition of Aβ amyloids is a hallmark of Alzheimer's disease[12]. It has been reported that HIV-1 induces Aβ accumulation in brain endothelial cells[16]. Brains from HIV-infected individuals show an increased frequency of Aβ amyloids and Alzheimer's amyloid plaques compared to age-matched uninfected individuals[17,18], and regions of HIV replication may colocalize with sites of amyloid accumulation[19]. It has been suggested that HIV-1 may promote the accumulation of Aβ in the brain by increasing its synthesis[20], decreasing Aβ degradation[21], or affecting its transport across the blood-brain barrier[22]. It has also been suggested that HIV may trigger the formation of α-synuclein amyloids[23], which play a key role in Parkinson's disease[24,25]. This agrees with evidence that HIV-infected individuals show a higher risk of developing Parkinson's disease[26]. Amyloid deposition and development of neurological disorders are strongly associated with aging[25] but the molecular interplay between these processes and HIV infection is poorly understood.

[1]Institute of Molecular Virology, Ulm University Medical Center, 89081 Ulm, Germany. [2]Department of Neurology, Ulm University, 89081 Ulm, Germany. [3]Laboratory of Neuropathology, Institute of Pathology, Center for Clinical Research at the University of Ulm, 89081 Ulm, Germany. [4]Laboratory of Neuropathology, Department of Imaging and Pathology, Leuven Brain Institute, KU Leuven, 3001 Leuven, Belgium. [5]Central Facility for Electron Microscopy, Ulm University, 89081 Ulm, Germany. [6]German Center for Neurodegenerative Diseases (DZNE), 89081 Ulm, Germany. [7]These authors contributed equally: Lia-Raluca Olari, Sichen Liu. ✉e-mail: frank.kirchhoff@uni-ulm.de

It has been established that amyloid fibrils formed by naturally occurring fragments of prostatic acid phosphatase and semenogelins, which are highly abundant in human semen, boost HIV attachment and infection[27–33]. Amyloid fibrils are readily detectable in semen samples from healthy individuals[34]. They may serve as "quality control" in reproduction by trapping damaged spermatozoa[35] and participate in innate immunity by binding bacteria, thereby protecting against infections from sexually transmitted bacterial pathogens[36]. Notably, peptides derived from the HIV-1 envelope (Env) glycoprotein gp120 may also form amyloid fibrils and promote infection[37,38]. These peptides, such as the 12-mer enhancing factor C (EF-C) corresponding to residues 417–428 of gp120, are highly effective in fibril formation and improve retroviral gene transfer[37]. Early studies have shown that Aβ42 fibrils bind HIV-1 particles and promote viral entry into human cell lines[39,40]. However, while the effects of semen and Env-derived amyloids on HIV infection of CD4+ T cells and macrophages are well established, it remains largely unclear whether amyloid fibrils associated with Alzheimer's and Parkinson's disease also affect HIV-1 infection and thus viral invasion of the brain and HIV-associated neu-

rological disorders. To address this, we analyzed the impact of Aβ and α-synuclein fibrils on the ability of HIV-1 to infect relevant human cell types including microglia representing the main target cells for HIV-1 infection in the brain[41,42]. We also examined whether gp120-derived amyloid EF-C fibrils promote aggregation of Aβ peptide and α-synuclein. Our results indicate that amyloids promote HIV-1 invasion of the brain and associated neurocognitive disorders.

## Results

### Aβ, α-synuclein and EF-C fibrils show distinct biophysical features

For structural and functional analyses, we chemically synthesized the EF-C (Enhancing factor C) peptide, while human Aβ40 peptide and α-synuclein protein were commercially acquired. The three amyloidogenic agents range in size from 12 (EF-C), 40 (Aβ40), to 140 (α-synuclein) amino acid residues (Fig. 1a). The EF-C peptide[37] that has been marketed as an enhancer of retroviral gene transfer under the brand name Protransduzin[43], has a positive charge and zeta potential, similar to semen-derived fibrils[27–29]. EF-C corresponds to residues 417–428 of

**a**

| Fibril | # of amino acids | Zeta potential (mV) | # of (+) charged residues | # of (-) charged residues | Theoretical pI |
|---|---|---|---|---|---|
| Aβ40 | 40 | -15 | 3 | 6 | 5.31 |
| | DAEFRHDSGYEVHHQKLVFFAEDVGSNKGAIIGLMVGGVV | | | | |
| α-synuclein | 140 | -35 | 15 | 24 | 4.67 |
| | MDVFMKGLSKAKEGVVAAAEKTKQGVAEAAGKTKEGVLYVGSKTKEGVVHGVATVAEKTKEQVTNVGGAVVTGVTAVAQKTVEGAGSIAAATGFVKKDQLGKNEEGAPQEGILEDMPVDPDNEAYEMPSEEGYQDYEPEA | | | | |
| EF-C (HIV-1 gp120, aa 417-428) | 12 | +25 | 2 | - | 9.31 |
| | QCKIKQIINMWQ | | | | |

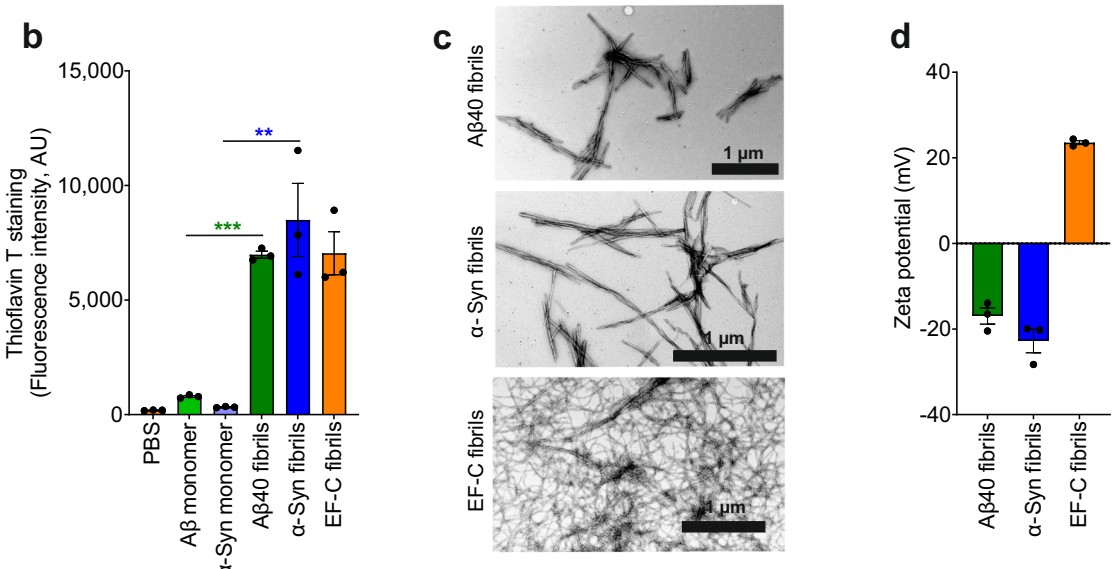

**b**  **c**  **d**

Fig. 1 | Aβ40 and α-synuclein form amyloid fibrils showing a negative zeta potential. a Table summarizing features of Aβ40, α-synuclein (α-Syn), and EF-C including the amino acid sequence analysis, zeta potential, and isoelectric point (pI) as calculated with the ProtParam tool. Highlight hydrophilic amino acids blue and hydrophobic amino acids red. b Monomeric and fibrillar peptides and proteins were incubated with Thioflavin T and fluorescence intensity was measured (arbitrary units, AU). Shown is the mean of three independent experiments measured in duplicates ± SEM. Source data are provided in the Source Data file. c TEM images of

fibrils. Samples were prepared by negative staining with 2% uranyl acetate in water on copper grids and imaged with a Jeol TEM 1400. Scale bars indicate 1 μm. Shown are representative images derived from one experiment out of three performed. d Zeta potential of fibrils diluted in ddH₂O. Samples were analyzed by nanoparticle tracking analysis. Shown are the means of three independent experiments each performed in duplicates ± SEM. Significant differences were determined using two-sided unpaired t-test analysis. Asterisks indicate statistical significance and exact P values in (b) were ***P < 0.0001 and **P = 0.0068.

the external HIV-1 envelope glycoprotein gp120[37]. Notably, related gp120-derived amyloidogenic peptides have been detected in the cerebrospinal fluid (CSF) of AIDS patients[44]. In contrast to EF-C and semen-derived fibrils, Aβ40 and α-synuclein contain an excess of negatively charged amino acids, and are hence predicted to have overall negative zeta potentials and lower isoelectric points compared to EF-C (Fig. 1a). While EF-C forms fibrils instantaneously in solution[37,45], Aβ40 and α-synuclein were agitated for several days, as described in the methods section. Agitated but not freshly dissolved Aβ40 and α-synuclein efficiently bound Thioflavin T amyloid dye (Fig. 1b). Transmission electron microscopy verified the efficient formation of fibrils showing a long and relatively straight morphology (Fig. 1c). Zeta potential measurements confirmed that Aβ40 and α-synuclein fibrils display a net negative surface charge, while EF-C fibrils are positively charged (Fig. 1d).

## α-synuclein fibrils enhance HIV-1 infection

We determined the effect of Aβ40, α-synuclein, and EF-C fibrils on four different HIV-1 strains. The CXCR4 (X4)-tropic infectious molecular clone (IMC) HIV-1 NL4-3 was used because it has been characterized in numerous previous studies. To assess the impact of the viral coreceptor tropism, we used an otherwise isogenic CCR5 (R5)-tropic NL4-3 derivative containing the Env V3-loop region of HIV-1 92th014.12[46]. Additionally, we used two HIV-1 strains originally derived from brain (JR-CSF[47]) and lung tissue (BAL[48]) that were previously shown to infect microglia[49]. To determine the impact of the three types of amyloids on viral infection, HIV-1 was produced by transfection of HEK293T cells and briefly pre-incubated with freshly dissolved peptides or their fibrillar forms. EF-C fibrils form instantaneously[37] precluding analysis of monomeric peptides. Thereafter, the mixtures were added to TZM-bl reporter cells and infection rates were determined three days later (Fig. 2a, left). The TZM-bl cell line was engineered to express the HIV-1 receptors CD4, CCR5, and CXCR4[50] and contains the reporter genes firefly luciferase and *E. coli* β-galactosidase under the control of an HIV-1 LTR[51]. Infection with HIV-1 containing the viral transactivator Tat activates the reporter genes, allowing highly sensitive and accurate measurements of infection. α-synuclein and EF-C fibrils dose-dependently enhanced infection of all four HIV-1 strains in TZM-bl reporter cells up to 100-fold, while Aβ40 fibrils had only modest effects, and monomeric peptides had no enhancing effects (Fig. 2a, Supplementary Figs. 1a, b). Fibrils alone did not induce reporter activity in TZM-bl cells (Fig. 2a). The enhancing effects on reporter gene expression and viral p24 production correlated very well (Supplementary Figs. 1c-e). Since the TZM-bl infection assay is highly sensitive and linear over a broad range, it was used in subsequent experiments. Heat treatment impaired the enhancing activity of α-synuclein but not of EF-C fibrils (Supplementary Fig. 1f). This agrees with data showing that some types of amyloids are highly resistant to heating[52]. Treatment with the entry inhibitors AMD3100 and Maraviroc efficiently inhibited X4- and R5-tropic HIV-1 infection, respectively, in both the absence and presence of amyloid fibrils (Supplementary Fig. 2). To verify the impact of the amyloids on HIV-1 infection in T cells, we used CEM-M7 cells, a T/B cell hybrid cell line that contains an LTR-driven GFP reporter gene cassette. Fluorescence-activated cell sorting (FACS) analyses confirmed that EF-C and (to a lesser extent) α-synuclein fibrils promoted HIV-1 infection (Fig. 2b, Supplementary Fig. 3). Of note, infection enhancement by α-synuclein was substantially higher than that of Aβ fibrils, with a maximum increase of ~60-fold in TZM-bl cells and ~10-fold in CEM-M7 cells compared to just up to 3-fold effects by Aβ amyloids.

## α-synuclein fibrils boost HIV-1 infection both directly and in trans

To determine whether Aβ40, α-synuclein and EF-C fibrils promote virus infection of cell types found in the brain, we examined their effect on HIV-1 infection in HMC-3 and U373-MAGI cells. The human microglial HMC-3 cell line was isolated from the brain of a patient and does not express CD4, or the HIV-1 coreceptors, CCR5 and CXCR4[53]. For infection, we used an R5-tropic HIV-1 NL4-3 construct expressing the Gaussia luciferase that is secreted into the cell culture supernatant and particularly suitable for monitoring infection kinetics. In line with previous data[54], HMC-3 cells were not susceptible to HIV-1 infection irrespectively of the absence or presence of amyloid fibrils (Fig. 2c). The human glioblastoma-astrocytoma cell line U373-MAGI was derived from a malignant brain tumor and expresses CD4 but not CCR5, and only minimal amounts of CXCR4[55]. Gaussia luciferase activity was readily detectable in the supernatant of U373-MAGI exposed to HIV-1 (Fig. 2c). However, Aβ40 had no significant enhancing effect and α-synuclein fibrils increased HIV-1 infection only at the highest dose. In comparison, EF-C fibrils increased HIV-1 LTR-driven Gaussia expression by up to 30-fold in a dose-dependent manner (Fig. 2c). Thus, attachment of HIV-1 particles to the cell surface by amyloid fibrils in the presence of CD4 may facilitate viral entry despite the absence of coreceptors. It has been shown that semen-derived fibrils enhance *trans*-HIV infection of T cells by dendritic or epithelial cells[28]. To assess whether amyloids may also promote *trans*-HIV infection in the brain, we exposed HMC-3 cells to a fixed amount of HIV-1 pretreated with different doses of the three types of amyloids. Subsequently, the unbound virus was removed by extensive washing, followed by co-cultivation with TZM-bl cells (Fig. 2d, left). Remarkably, Aβ40, α-synuclein, and EF-C fibrils all increased the ability of HMC-3 cells to transfer HIV-1 to TZM-bl cells in a dose-dependent manner (Fig. 2d). In agreement with the effects on cell-free virus infection, Aβ40 showed the lowest efficiency, while α-synuclein and EF-C fibrils increased in trans-HIV-1 infection by up to two orders of magnitude.

## α-synuclein fibrils enhance infection of primary HIV-1 target cells

To assess the potential in vivo relevance, we examined the effect of Aβ, α-synuclein, and EF-C fibrils on infection of primary CD4+ T cells, the major HIV-1 target cells in infected individuals. CD4+ T cells can cross the blood-brain barrier and contribute to seeding and maintaining HIV-1 infection in the brain, as well as to inflammation and the development of neurological symptoms[56]. Activated primary CD4+ T cells were exposed to HIV-1 either untreated or treated with different amounts of amyloid fibrils. The culture supernatants were collected on different days post-infection and examined for the presence of infectious virus particles by TZM-bl reporter assay (Fig. 3a, left). Treatment with α-synuclein and EF-C accelerated and increased the production of infectious HIV-1 R5- and X4-tropic NL4-3, as well as AD8 (a macrophage-tropic IMC[57]) and JR-CSF (Fig. 3a). Consistent with results obtained using cell lines (Fig. 2), Aβ amyloid had only modest effects at the highest dose. Wild-type X4-tropic HIV-1 NL4-3 replicated efficiently in CD4+ T cells in the absence of amyloid treatment. In contrast, untreated brain-derived HIV-1 JR-CSF showed only low basal levels of replication that were increased by up to two orders of magnitude by the treatment with α-synuclein and EF-C fibrils (Fig. 3a).

The role of macrophages in HIV-1 invasion of the brain and as a source for brain inflammation and HAND is well known[58]. To determine whether amyloids enhance HIV-1 infection of macrophages, monocyte-derived macrophages (MDMs) were generated from human peripheral blood mononuclear cells (PBMCs) by culturing them for 10 days in a medium containing human serum and M-CSF. We found that EF-C fibrils substantially increased replication of R5-tropic HIV-1 NL4-3 and AD8 in macrophages, while Aβ fibrils had little effect, and α-synuclein fibrils resulted in intermediate levels of viral replication (Fig. 3b). In contrast, X4-tropic NL4-3 did not replicate above background levels in macrophages.

Microglia represent the main target cells for HIV-1 replication in the brain[41] and serve as persistent viral reservoirs during long-term

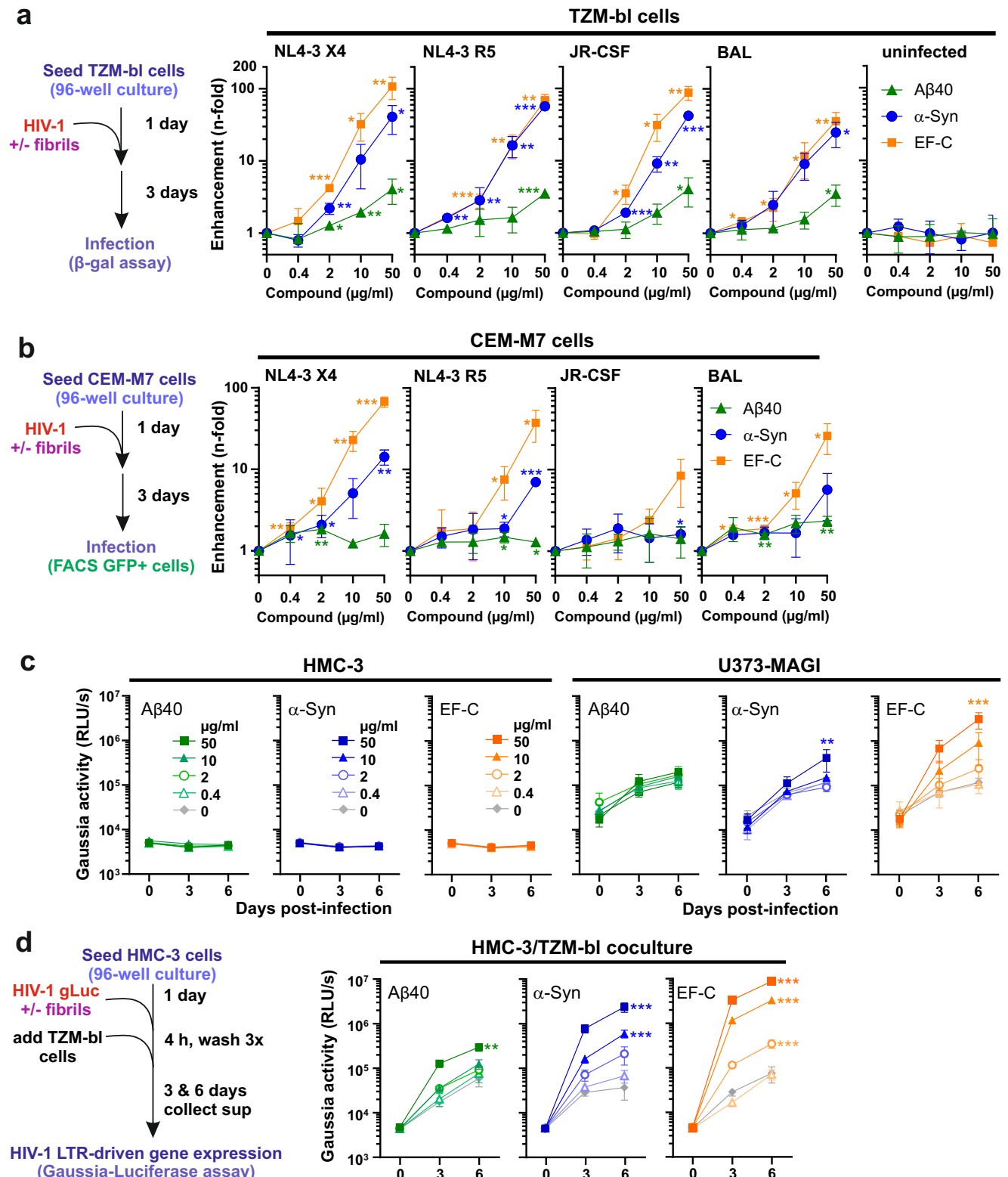

combined antiretroviral therapy (cART)[59]. To study their susceptibility to HIV-1 infection and the impact of amyloids, we generated monocyte-derived microglia cells (MMGs) from human PBMCs as previously reported[60]. MDMs and MMGs were examined for the expression of specific markers (CD45, IBA1, and P2RY12), as well as HIV-1 entry factors (CD4, CCR5, and CXCR4). Predictably, macrophages and microglia expressed both CD45 and IBA1, while P2RY12 was only expressed by microglia (Supplementary Fig. 4a, b). Microglia expressed higher levels

of CD4 and both viral coreceptors compared to macrophages (Supplementary Fig. 4c). Unlike macrophages, they allowed significant replication of X4-tropic HIV-1 upon treatment with EF-C or α-synuclein fibrils (Fig. 3c). In addition, microglia produced ~5-fold higher levels of infectious HIV-1 AD8 than macrophages. Despite extensive washing, microglia showed relatively high background levels of infectious HIV-1 early after virus exposure. Most likely, this represents input virus and suggests that virion/fibril aggregates efficiently bind to microglia but a

**Fig. 2 | Aβ40, α-synuclein, and EF-C fibrils enhance HIV-1 infection. a, b** Aβ40, α-synuclein (α-Syn), and EF-C fibrils were pre-incubated with different HIV-1 strains and added to (**a**) TZM-bl and (**b**) CEM-M7 cells. Infection was quantified three days post-infection by detecting the expression of β-galactosidase in the TZM-bl cells or quantifying the GFP + CEM-M7 cells by flow cytometry. Values were corrected for the background signal derived from the uninfected cells, and infection efficiencies are provided as *n*-fold changes relative to those observed in the absence of fibrils (1×). **c** Aβ40, α-synuclein, and EF-C fibrils were pre-incubated with a Gaussia luciferase HIV-1 reporter virus and added to HMC-3 (left) or U373-MAGI cells (right).

**d** In trans-infection of TZM-bl by fibril-virus mixtures preincubated with HMC-3 cells for 4 h, followed by extensive PBS washing, and the addition of TZM-bl cells. Virus replication was measured by testing the Gaussia luciferase activity (relative light units per second, RLU/s) in supernatants harvested at the indicated time points. In all cases, the compound concentration during virion pre-incubation is indicated. Shown is the mean of three independent experiments measured in triplicates ± SEM. Significant differences were determined using two-sided unpaired t-test analysis. Asterisks indicate statistical significance (\**P* < 0.05, \*\**P* < 0.01, \*\*\**P* < 0.001). Source data are provided as a Source Data file.

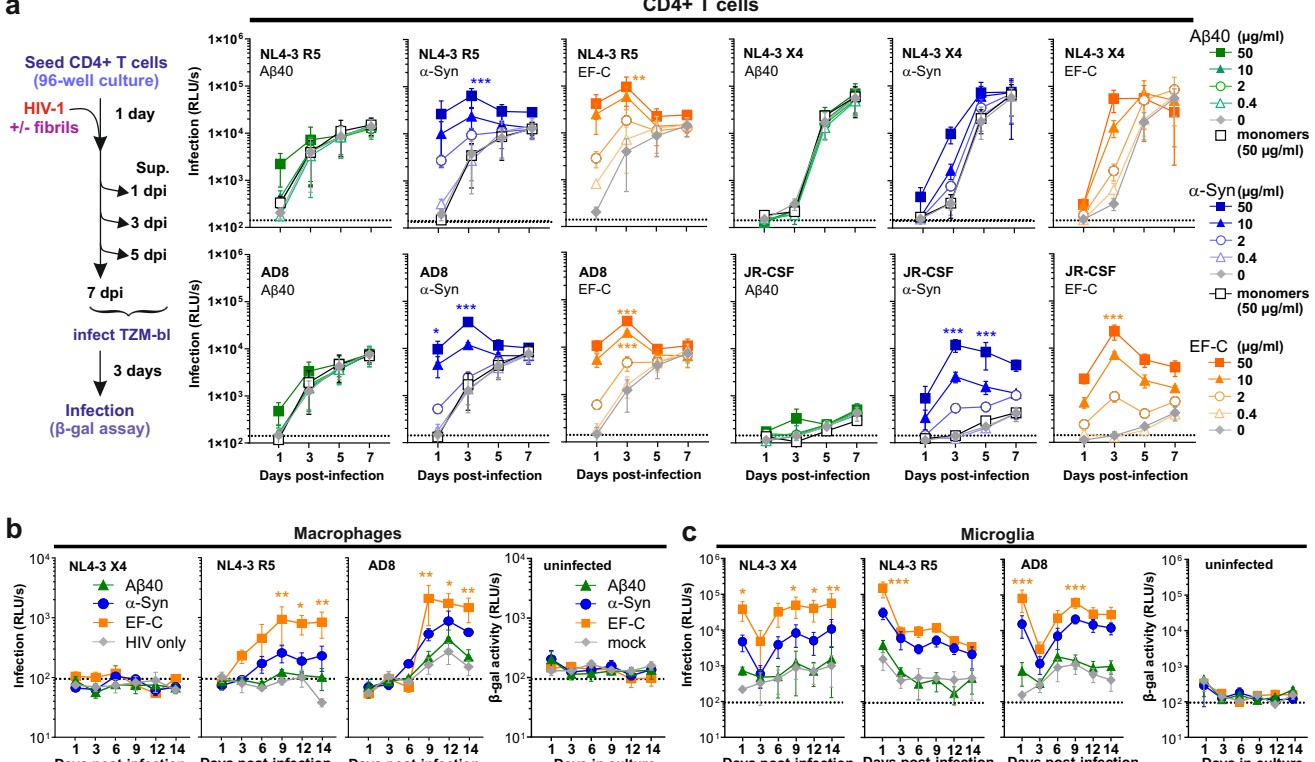

**Fig. 3 | Aβ40, α-synuclein, and EF-C fibrils enhance HIV-1 infection and replication in primary cells.** Aβ40, α-synuclein (α-Syn), and EF-C fibrils at indicated concentrations were pre-incubated with different HIV-1 strains (or cell medium as control) and added to (**a**) CD4⁺ T cells, PBMC-derived (**b**) macrophages and (**c**) microglia. One day post-infection, cells were extensively washed in PBS to remove the input virus. Supernatants were harvested at the indicated time points and added to the TZM-bl reporter cell line. Infection was quantified three days post-infection by detecting the expression of β-galactosidase (relative light units per second, RLU/s). Dashed lines indicate the background signal derived from the uninfected cells. Shown is the mean of three independent experiments using cells derived from different donors, measured in triplicates ± SEM. Significant differences were determined using two-sided unpaired t-test analysis. Asterisks indicate statistical significance (\**P* < 0.05, \*\**P* < 0.01, \*\*\**P* < 0.001). Source data are provided as a Source Data file.

significant proportion does not fuse and is released over time. However, infectious titers of HIV-1 increased again at later time points indicating productive replication (Fig. 3c). In contrast, only marginal infectious virus production was detected after exposure to HIV-1 that was left untreated or treated with Aβ40 fibrils. These results suggest that α-synuclein amyloid may strongly increase the spread of HIV-1 in brain macrophages and microglia.

### α-synuclein fibrils promote HIV-1 virion fusion

It has been previously established that semen-derived fibrils enhance the attachment and fusion of HIV-1 particles[28,29,61]. To determine whether Aβ and α-synuclein amyloids also promote virion fusion, macrophages and microglia were infected with BlaM-Vpr-containing HIV-1 AD8 virions pretreated with 50 μg/mL of the indicated amyloid fibrils or left untreated (Fig. 4a). Transfer of the viral capsid into the target cells upon fusion is monitored by enzymatic cleavage of CCF2 by β-lactamase, which shifts the

fluorescence emission spectrum of the dye from green (520 nm) to blue (447 nm)[62]. The number of infected blue cells markedly increased upon treatment with α-synuclein and (most strongly) EF-C fibrils, while Aβ40 fibrils had little enhancing effect in macrophages and no effect in microglial cells (Fig. 4b). Both α-synuclein and EF-C fibrils significantly enhanced the fusion of HIV-1 particles with macrophages and microglia (Fig. 4c) and most of the latter were infected upon virion treatment with EF-C (Fig. 4b). These results support that α-synuclein and EF-C amyloids enhance HIV-1 replication in macrophages and microglia by increasing viral entry.

### α-synuclein and EF-C fibrils promote HIV-1 attachment to target cells

The positive surface charge of semen-derived fibrils promotes HIV-1 attachment by counteracting the repulsion between the negatively charged viral and cellular membranes[27,61]. Semen-derived

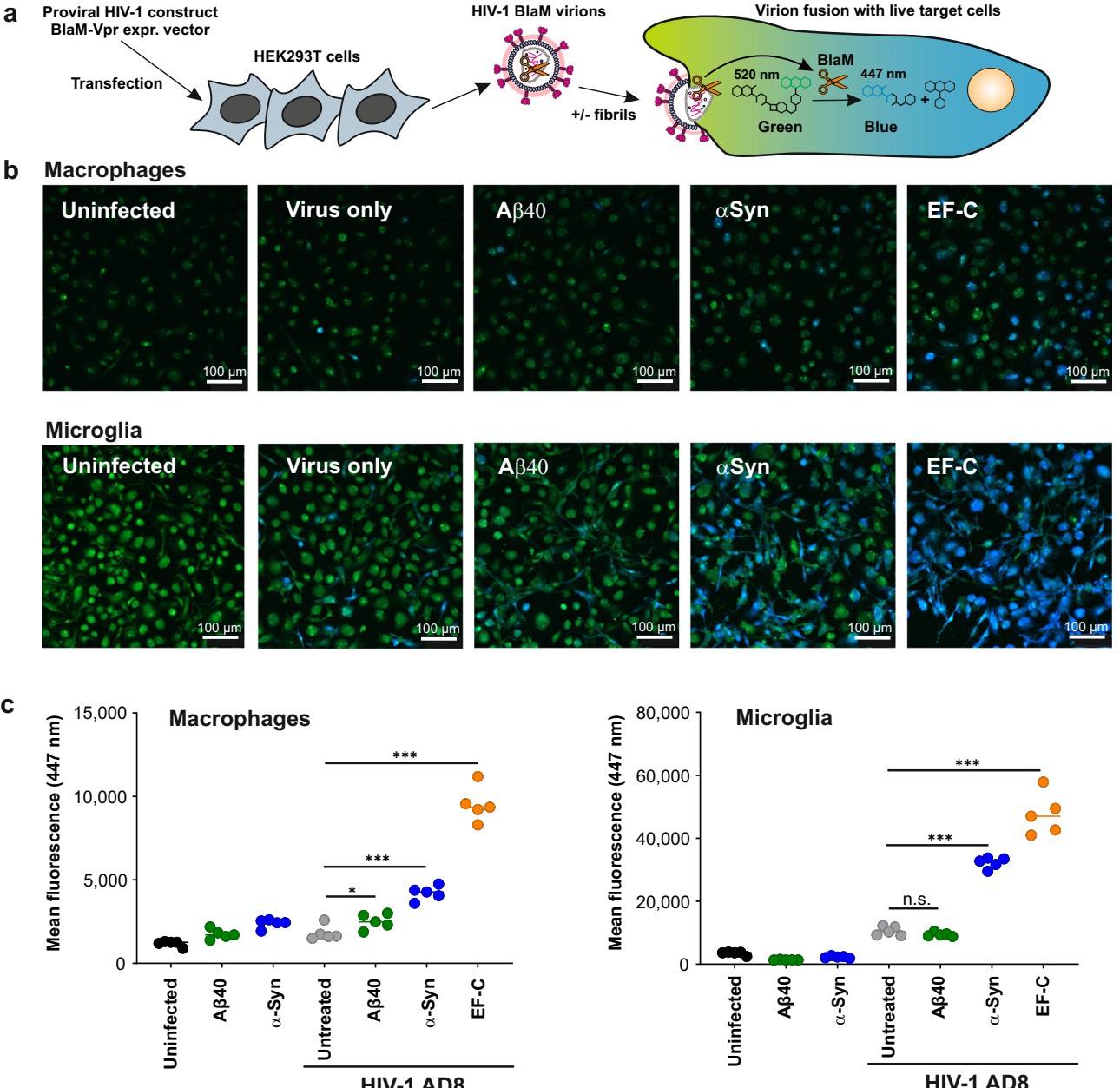

**Fig. 4 | Aβ40 and α-synuclein fibrils enhance HIV-1 fusion in PBMC-derived macrophages and microglia. a** Schematic of HIV-1 BlaM-Vpr fusion assay. HIV-1 AD8 proviral DNA and pCMV-BlaM-Vpr expression vector were cotransfected into HEK293T cells to generate HIV-1 BlaM-Vpr virions. These virions were used to infect target cells in the presence or absence of fibrils. Fusion with target cells was monitored by a fluorescence shift from green (520 nm) to blue (447 nm) upon cleavage of the CCF2 substrate by BlaM-Vpr, indicating viral entry. **b** Macrophages and microglia cells were mock-infected or infected with HIV-1 virions containing BlaM-Vpr with or without being mixed 1:1 with final 50 μg/mL concentration (final concentration on virus) Aβ40 (Aβ), α-synuclein (αSyn), and EF-C fibrils. Cells were loaded with CCF2 dye and analyzed by fluorescence microscopy. The images correspond to the 530 and 460 nm emission after a fluorophore excitation at 405 nm. Scale bars indicate 100 μm. Images were taken with a Leica D8i confocal microscope (Leica). **c** Quantification of the mean fluorescence of the 460 nm channel in macrophages and microglia cells. Data represent the average fluorescence from five different images per condition. Shown is one representative experiment. Highly similar results were obtained in two additional independent experiments. 'Source data are provided as a Source Data file. Significant differences were determined using one-way ANOVA with Dunnett's multiple comparison test. Asterisks indicate statistical significance (*$P = 0.0396$, ***$P < 0.0001$).

amyloid fibrils closely interact with both viral and cellular membranes and increase virion attachment[27–29]. To investigate whether neuronal amyloids may act by a similar mechanism, we analyzed by confocal microscopy whether fibrils stained with the Amytracker amyloid dye bind CFP-labeled HIV-1 particles and promote attachment. We found that complexes between HIV-1 particles and amyloids are readily detectable at the surface of TZM-bl cells (Fig. 5a, Supplementary Fig. S5). In agreement with the results obtained in HIV-1 infection experiments, and in line with this, Aβ and α-

synuclein fibrils also increased the number of murine leukemia virus (MLV) particles in close proximity to TZM-bl cells (Supplementary Fig. 6). Altogether, the results showed that fluorescently stained Aβ, α-synuclein and EF-C amyloids efficiently and rapidly capture retroviral particles, interact with TZM-bl cells, and increase the number of virions detectable at the cell surface. Thus, similar to semen-derived fibrils, Aβ40 and α-synuclein fibrils interact with viral and cellular membranes despite their overall negative surface charge.

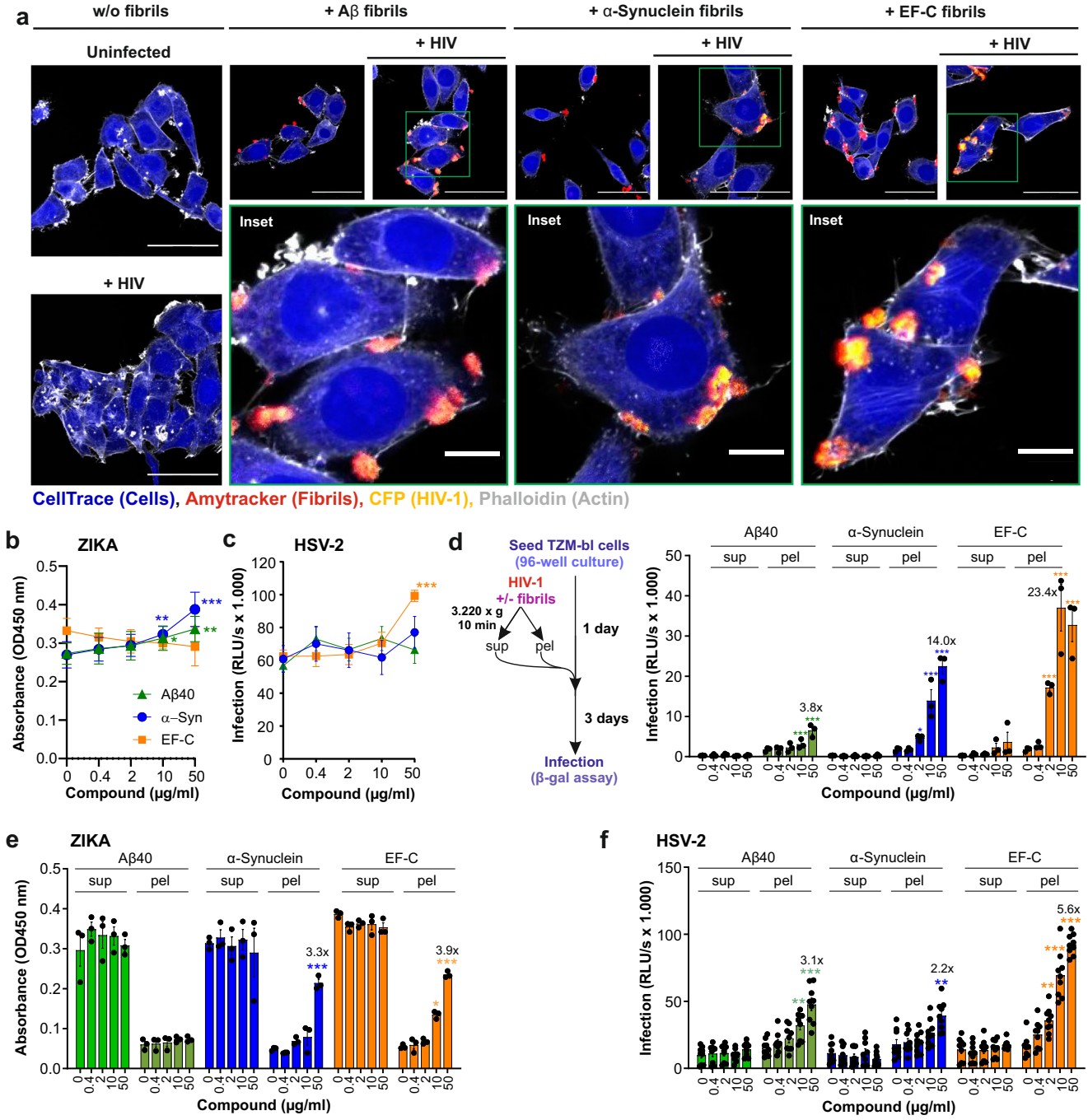

**Fig. 5 | Aβ40/42 and α-synuclein fibrils enhance HIV infection and attach retroviral particles to the target cells. a** Fluorescence microscopy images showing Aβ40, α-synuclein, and EF-C fibrils that were stained with Amytracker 540 dye (red) and CFP labeled HIV (yellow) in the absence and presence of cells stained with the cytoskeleton dye (CellTrace) (blue) and actin dye (ATTO-phalloidin) (white). Scale bars are 50 μm for the full images and 10 μm for the insets. Images were taken with an LSM 710 confocal microscope (Zeiss). Shown are representative images derived from one experiment. **b**–**d** Aβ40, α-synuclein (α-Syn), and EF-C fibrils were pre-incubated with (**b**) ZIKV or (**c**) HSV-2 and added to (**c**) Vero E6 or (**d**) ELVIS cells. Infection was quantified three days post-infection via ZIKV ELISA or one day post-infection by detecting the expression of β-galactosidase (relative light units per second, RLU/s) in the ELVIS cells. HIV-1, ZIKV or HSV-2 were incubated with the indicated concentrations of Aβ40, α-synuclein, and EF-C fibrils. The supernatant and pellet were separated by centrifugation and the pellet was resuspended in fresh medium. Samples were added to (**d**) TZM-bl cells, (**e**) Vero E6 cells, or (**f**) ELVIS cells. The infection was quantified one (**f**), two (**e**), or three (**d**) days post-infection by (**d**, **f**) detecting the expression of β-galactosidase (relative light units per second, RLU/s) or (**e**) via ZIKV ELISA. Values were corrected for the background signal derived from the uninfected cells. Shown is the mean of three independent experiments measured in triplicates ± SEM. Significant differences were determined using two-sided unpaired t-test analysis and two-way ANOVA with Dunnett´s multiple comparison test. Asterisks indicate statistical significance (*$P < 0.05$, **$P < 0.01$, ***$P < 0.001$). Source data are provided as a Source Data file.

## α-synuclein fibrils enhance HIV-1 but not Zika virus and HSV-2 infection

To assess the specificity of the infection enhancement, we tested the impact of the three types of amyloids on Zika virus (ZIKV) and Herpes simplex virus type 2 (HSV-2) infection and interaction. Both viruses are known to invade the brain and cause neurological complications[63,64]. Pretreatment with amyloid fibrils did not enhance ZIKV or HSV-2 infection of Vero E6 cells[65] and ELVIS β-galactosidase reporter cells[66],

respectively (Fig. 5b, c). It has been shown that the interaction between EF-C and viral particles allows for the rapid concentration of infectious HIV-1 by low-speed centrifugation[37]. Similar to EF-C, α-synuclein fibrils efficiently pelleted infectious virus and enhanced HIV-1 infection in a dose-dependent manner, while Aβ had only modest effects (Fig. 5d). While α-synuclein and EF-C pelleted HIV-1 and increased infection (Fig. 5d), most infectious ZIKV remained in the supernatant (Fig. 5e). HSV-2 was detectable in both the supernatant and the pellet with increasing quantities of amyloid fibrils moderately enhancing infection efficiencies (Fig. 5f). Altogether, these results indicate that α-synuclein fibrils, similar to EF-C, bind HIV-1 and promote infection, but have little, if any effect on ZIKV and HSV-2 infection.

### EF-C fibrils accelerate the amyloid formation of α-synuclein

It has been reported that HIV-1 gp120-derived fibrils, such as EF-C, promote the formation of SEVI (semen-derived enhancer of virus infection) fibrils composed of residues 248–286 of prostatic acid phosphatase[67]. A variety of gp120-derived peptides that are prone to form amyloid fibrils and have been detected in the CSF of AIDS patients[44]. To assess whether amyloidogenic HIV-1 gp120-derived peptides might also promote amyloid formation in the brain, we mixed monomeric Aβ40 or α-synuclein with low proportions of HIV-1 Env-derived EF-C peptide amyloid seeds. We observed slightly accelerated aggregation when Aβ40 was mixed with 10% EF-C (Fig. 6a). In comparison, α-synuclein aggregated much faster and more efficiently in the presence of 10% EF-C seeds (Fig. 6b). Aggregation was also observed in the presence of 1% EF-C but not for the unseeded α-synuclein sample within the 153-h time frame investigated. To examine whether HIV-1 virions may impact the formation of brain-derived fibrils, we incubated the Aβ40 peptide with sucrose-purified viral particles. Purified HIV-1 particles did not accelerate the formation of Aβ40 fibrils (Fig. 6c). To specifically visualize the assembly of 10% EF-C cross-seeded Aβ40 and α-synuclein fibrils, 2% (v/v) of the peptide or protein monomers were labeled with different ATTO dyes prior to the initiation of fibril formation. Microscopic analyses showed that non-seeded fibrils displayed fluorescence only in the channel of the ATTO dye used. Upon addition of EF-C seeds, mixed fibrils were observed, in which the seeds and Aβ40 or α-synuclein were homogenously distributed throughout the aggregates (Fig. 6d). Notably, mixed cross-seeded fibrils enhanced HIV-1 infection as efficiently as their homogeneous counterparts (Fig. 6e; Supplementary Fig. 7). In contrast, monomeric Aβ40 and α-synuclein, as well as the EF-C seed controls had minimal to no effect on HIV-1 infection. Altogether, these results show that HIV-1-Env derived amyloidogenic peptides have the potential to promote amyloid formation by α-synuclein and (less efficiently) Aβ40 and further indicate that the heterogeneous nature of these fibrils does not impact their ability to interact with HIV-1 particles.

### Cell-secreted α-synuclein promotes HIV-1 infection

Cell-free α-synuclein fibrils potently enhanced HIV-1 infection. However, α-synuclein is expressed in the cells and it remained to be determined whether cell-secreted α-synuclein has similar effects. To investigate this, we transfected H4 neuroglioma cells with α-synuclein constructs containing a split Gaussia luciferase complementation system, that is indicative for α-synuclein oligomerization or the formation of higher order structures and allows quantification of oligomers released by the cells[68]. In addition to wild type α-synuclein oligomers, we also analyzed the effects of oligomers formed by α-synuclein variants with nonsynonymous point mutations associated with autosomal dominant forms of Parkinson's disease, which are reported to promote dimerization (A30P, E46K, A53T)[69]. We found that secreted α-synuclein oligomers and/or aggregates dose-dependently enhance HIV-1 infection in TZM-bl cells (Fig. 7; Supplementary Fig. 8). Depletion of α-synuclein by antibodies abolished the enhancing effect and it was not observed using supernatants of cells

transfected with an empty control vector. The presence of point mutations in α-synuclein did not impact the enhancing effect on HIV-1 infection (Fig. 7). Our finding that α-synuclein oligomers released from cells promote HIV-1 infection suggests that enhancement may already occur at early stages of synucleinopathies.

### Human brain lysates enhance HIV-1 infection

To assess whether primary amyloids derived from human tissue enhance HIV-1 infection, we examined human brain lysates from elderly individuals including three patients with Alzheimer's and/or Lewy body disease (LBD), and three individuals without documented neurological complications (Supplementary Table 1). Brain lysates derived from Alzheimer's patients have been described to contain high molecular weight (>1000 kDa) Aβ species (protofibrils and fibrils) in the soluble fraction[70]. To produce the lysates, brain tissue was homogenized, diluted in PBS, and centrifuged (14,000 × g) to pellet the insoluble compounds. The supernatants (i.e., lysates) were collected and normalized for total protein content. All samples showed binding to Thioflavin T, albeit at varying levels, suggesting differences in amyloid content among the brain lysates (Fig. 8a). To determine their effect on virus infection, R5-tropic HIV-1 NL4-3 was pre-treated with the lysates and added to TZM-bl cells (Fig. 8b). At the highest concentration (50% lysate), all brain samples enhanced HIV-1 infection by 7- to 11-fold (Fig. 8b). At a concentration of 10% the enhancing activity varied strongly and correlated with the signal measured in the Thioflavin T binding assay (Fig. 8c), suggesting that aggregated amyloid structures mediate the enhancing effect on HIV-1 infection. Treatment with brain extracts alone did not induce reporter gene expression in TZM-bl cells (Fig. 8d). In this small pilot study, brain lysates from the three patients with Alzheimer's and/or LBD were not more active in enhancing HIV infection than those from the remaining three individuals. Altogether, these preliminary results suggest that native Thioflavin T stainable aggregates derived from human brains enhance HIV-1 infection.

## Discussion

HIV-1 invades the brain in a significant number of infected individuals and accelerates the development of neurological disorders even in the absence of detectable plasma viremia[13,71,72]. Although HAND affects up to 60% of all people living with HIV and is thus of high clinical significance[73,74], the underlying mechanisms are poorly understood. Here, we show that amyloid fibrils formed by α-synuclein and (to a much lesser extent) Aβ peptide, which are associated with neurological disorders such as Parkinson's and Alzheimer's disease, respectively, boost HIV-1 infection of primary human T cells, macrophages, and microglia. Conversely, an amyloidogenic peptide derived from the HIV-1 Env protein (EF-C) cross-seeded and accelerated amyloid formation of α-synuclein and, less effectively, Aβ peptide. We further show that supernatants of α-synuclein-producing cells also enhance HIV-1 infection. Finally, extracts from human brains promoted HIV-1 infection, and the enhancing effect correlated with the levels of Thioflavin T-stainable material. However, it remains to be determined which types of amyloids were present and definitive conclusions about the roles of Aβ and α-synuclein fibrils in HAND will require larger studies in humans. Altogether, our results suggest that HIV-1 and brain amyloids may engage in a detrimental interplay that accelerates the development of neurological disorders.

It has been shown that a positive charge is critical for the ability of semen-derived amyloid fibrils to boost HIV-1 attachment and entry[61]. This is conceivable since HIV-1 particles contain only a few Env trimers and most contacts between virions and cells do not result in infection because the negatively charged viral and cellular membranes repel one another. Thus, semen-derived fibrils bind to both HIV virions and target cells most likely serving as a cationic bridge that facilitates virion attachment and fusion similarly to synthetic cationic polymers[75]. We found that α-synuclein amyloids and to a lesser extent Aβ fibrils

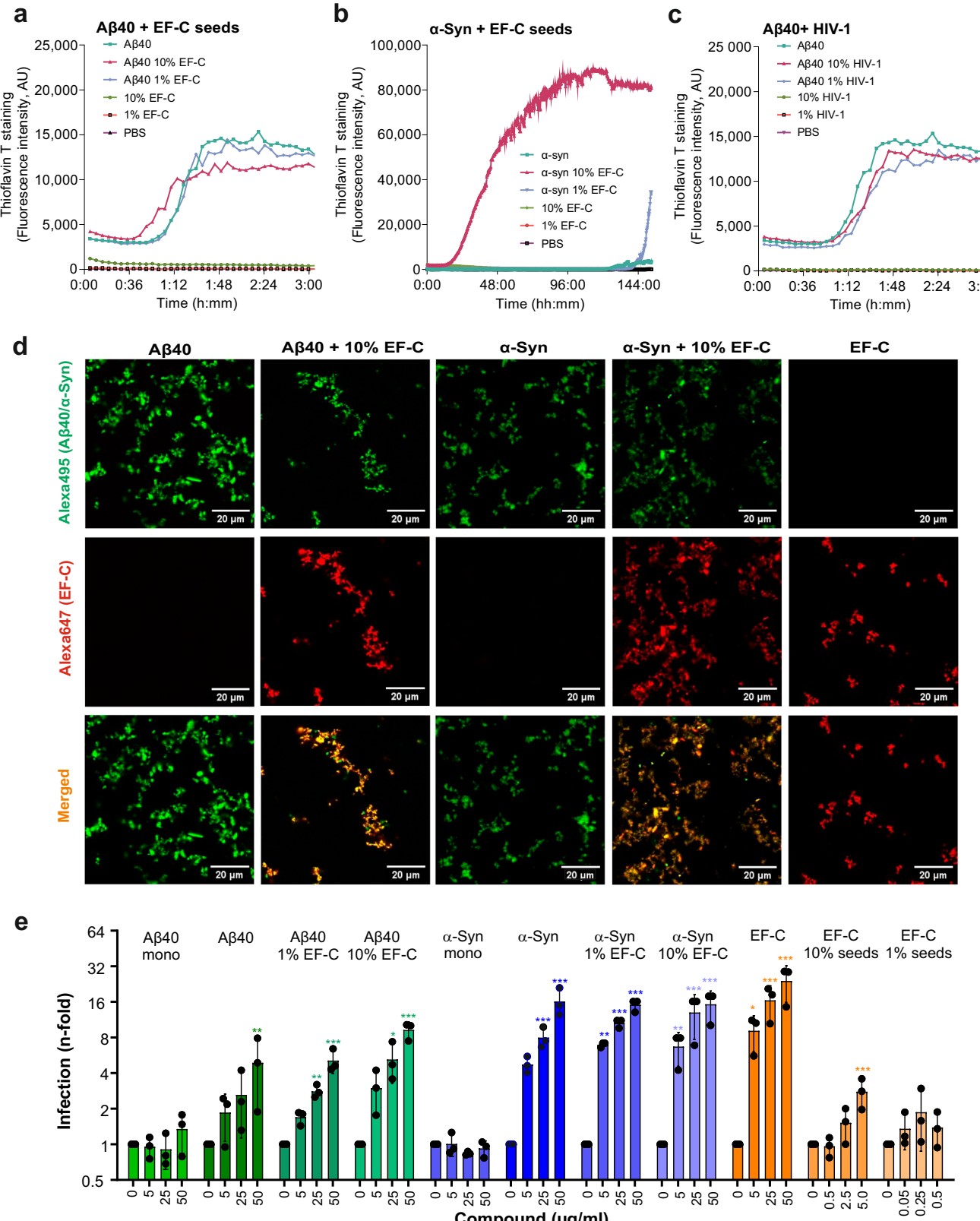

interact with virions and enhance infection despite their overall negative surface charge. However, membrane interactions via both electrostatic and hydrophobic interactions of Aβ and α-synuclein are well established[76] and are thought to play roles in the misfolding, aggregation, and toxicity of α-synuclein[77,78]. Notably, Aβ and α-synuclein bind particularly efficiently to small vesicles with high

surface curvature and lipid packing defects[79–82]. Viral particles share these features of extracellular vesicles[83]. However, binding to a single membrane might not be sufficient for infection enhancement since freshly dissolved Aβ and α-synuclein did not enhance HIV-1 infection. It has been reported that α-synuclein oligomers contain multivalent membrane binding sites and promote the clustering of vesicles and

**Fig. 6 | Aβ40 and α-synuclein fibrils are cross-seeded by HIV-1 Env-derived EF-C peptide.** Aggregation kinetics of (**a**) Aβ40 and (**b**) α-synuclein (α-Syn) cross-seeded with indicated amounts of EF-C seeds, or (**c**) Aβ40 cross-seeded with indicated amounts of sucrose-purified HIV-1 incubated with Thioflavin T (arbitrary units, AU) under agitation at 37 °C. Shown is the mean of one experiment measured in quadruplicates. **d** Confocal microscopy analysis of ATTO495-labeled Aβ40 and α-synuclein (α-Syn) fibrils cross-seeded with 10% ATTO647N EF-C. Scale bars indicate 20 μm. Images taken with a Leica D8i confocal microscope (Leica). Shown are representative images derived from one experiment. **e** Indicated pre-formed fibrils, seeds, monomers (mono) were pre-incubated with HIV-1 NL4-3 R5 tropic virus and

added to TZM-bl cells. Infection was quantified three days post-infection by detecting the expression of β-galactosidase. Values were corrected for the background signal derived from the uninfected cells, and infection efficiencies are provided as *n*-fold changes relative to those observed in the absence of peptide (1×). The compound concentration during virion pre-incubation is indicated. Shown is the mean of three independent experiments measured in triplicates ± SEM. Significant differences were determined using two-sided unpaired t-test analysis. Asterisks indicate statistical significance (*P < 0.05, **P < 0.01, ***P < 0.001). Source data are provided as a Source Data file.

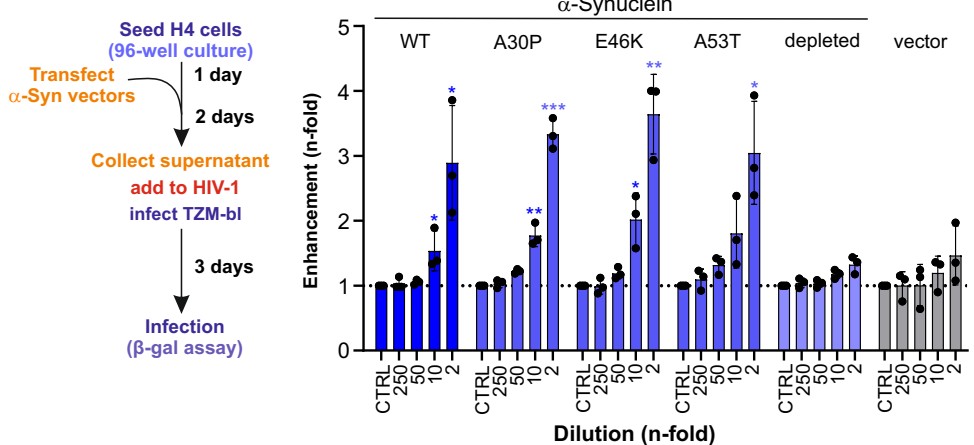

**Fig. 7 | α-synuclein enhances HIV-1 infection.** Cell supernatants containing α-synuclein oligomers were generated upon transfection of H4 cells with indicated α-synuclein constructs containing a split Gaussia luciferase construct, or with an empty vector as a control. Oligomer presence was quantified by Gaussia luciferase activity. α-synuclein was depleted in the supernatant of wild-type (WT) transfected H4 cells using magnetic protein G beads and SYN-1 antibody as described in the method section. Supernatants were diluted in PBS, and pre-incubated with HIV-1 AD8, and added to TZM-bl cells. Infection was quantified three days post-infection by detecting the expression of β-galactosidase in the TZM-bl cells. Values were corrected for the background signal derived from the uninfected cells, and infection efficiencies are provided as *n*-fold changes relative to those observed in the absence of peptide (1×). Shown is the mean of three independent experiments measured in triplicates ± SEM. Significant differences were determined using two-way ANOVA with Dunnett´s multiple comparison test. Asterisks indicate statistical significance (*P < 0.05, **P < 0.01, ***P < 0.001). Exact *P*-values are: 0.0391; 0.0206; 0.0015; <0,0001; 0.0122; 0.0017 and 0.0112 (from left to right). Source data are provided as a Source Data file.

hemifusion of negatively charged membranes[84,85]. It is thus tempting to speculate that multivalent α-synuclein oligomers and higher-order aggregates, capable of binding both HIV-1 particles and target cells, enhance infection almost as efficiently as positively charged EF-C fibrils. However, viral interactions with amyloids are complex[86,87], and recent data show that β sheet content and hydrophobic surfaces of amyloids also play a relevant role in HIV infection[88]. Further studies are required to fully clarify why α-synuclein fibrils are more effective than Aβ fibrils. Effects will also depend on the accessibility of the viral membrane for amyloid interaction and the intrinsic ability of the virus for attachment and entry into their respective target cells. Thus, the low number of Env trimers on HIV-1 particles[89] helps to explain why the enhancing impact of the fibrils is stronger on HIV-1 compared to neurotropic ZIKV and HSV-2.

Early analyses indicated that amyloid fibrils do not circumvent the requirement for CD4 and a coreceptor for HIV-1 infection[28]. However, by mediating virion attachment to the cell surface, amyloids reduce the threshold levels of these receptors required for viral infection. Microglia and macrophages, which represent major HIV-1 target cells in the brain, express lower levels of CD4 than T cells. To adapt to this CD4 low environment, the Env proteins of brain-derived HIV-1 strains typically show high affinity for the CD4 receptor[90,91]. In addition, a significant proportion of neurotropic HIV-1 strains lack the ATG initiation codon of the *vpu* gene. This increases Env expression since both are expressed from the same bicistronic mRNA[92]. In addition, we found that EF-C and to some extent α-synuclein fibrils allowed R5-

tropic HIV-1 to infect U373-MAGI cells although they do not express CCR5 at detectable levels. Thus, amyloid attachment factors lowering the threshold of CD4 and coreceptors required for efficient viral entry might be particularly important for facilitating HIV-1 infection and spread in the brain.

About 10 to 15% of brain cells are microglia[93,94]. Only a subset of them expresses the CD4 receptor. However, compelling evidence supports that microglia are the main target cells for productive HIV-1 infection in the brain[41,59]. Notably, fibrils also promote in trans infection of HIV-1 from cells that are not susceptible to infection to susceptible cells. This effect is likely mediated by virion/fibril complexes bound to the cell surface in a largely unspecific manner. Unlike macrophages and T cells, however, washing of the cells did not fully remove HIV-1 virions bound to microglia in the presence of fibrils. Instead, infectious input virus was released from microglia after exposure. Microglia are highly migratory and interact with one another and with other cell types[95]. Thus, while the physiological relevance of binding of virus-fibril aggregates to non-susceptible cells remains to be determined, it is tempting to speculate that CD4-negative microglia carrying fibril-virion complexes may spread HIV-1 in the brain by boosting in trans infection of CD4+ microglia and T cells, or macrophages. Spreading of HIV-1 by microglia might be particularly efficient under inflammatory conditions when they become activated and show high migratory activity[96].

We found that HIV-1 Env-derived EF-C fibrils can cross-seed and accelerate the formation of α-synuclein and to a lesser extent Aβ

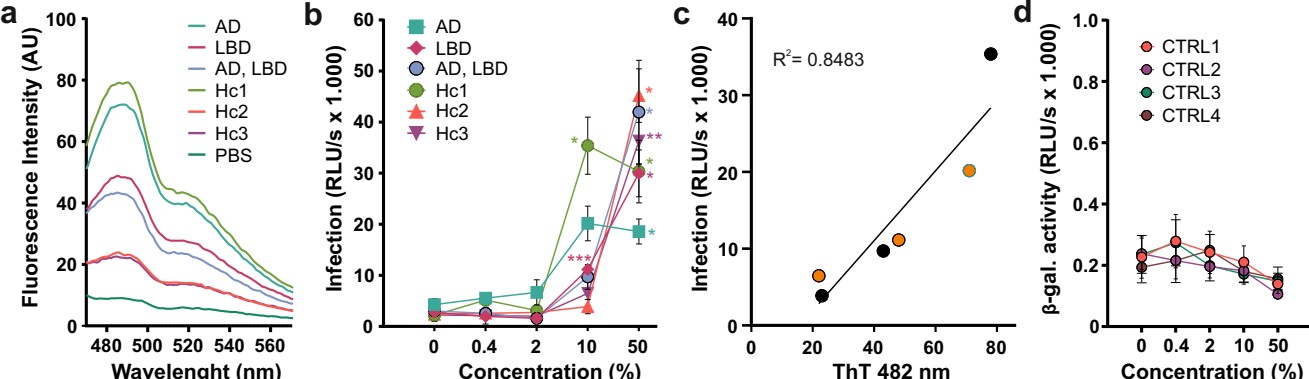

**Fig. 8 | Enhancement of HIV-1 infection by human brain lysates correlates with Thioflavin T binding. a** Quantification of the amyloid content of the soluble fraction of human brain lysates derived from Alzheimer's disease (AD), Lewy body disease (LBD) patients or individuals without documented neurological symptoms (Hc, healthy controls) by Thioflavin T staining and measurement of the fluorescence intensity. **b, d** The indicated concentrations of human brain lysates were pre-incubated with (**b**) HIV-1 NL4-3 R5 tropic or (**d**) cell medium and added to TZM-bl cells. Infection was quantified three days post-infection by detecting the expression of β-galactosidase (relative light units per second, RLU/s) in the TZM-bl cells. Shown is the mean of (**b**) one experiment measured in triplicates ± SD, or (**d**) three independent experiments measured in triplicates ± SEM. **c** Linear regression between the HIV-1 enhancing effect of 10% extract (v/v) on virus and the fluorescence intensity of the samples at 482 nm in the Thioflavin T assay. $R^2$ is the coefficient of determination. Color coding corresponds to that used in panel (**c**). Significant differences were determined using two-way ANOVA with Dunnett's multiple comparison test. Asterisks indicate statistical significance (*$P < 0.05$, **$P < 0.01$, ***$P < 0.001$). Source data are provided as a Source Data file.

fibrils. This agrees with previous studies showing that Aβ can cross-seed SEVI fibrils[97] and that EF-C promotes the formation of amyloid fibrils by another semen-derived amyloidogenic peptide (PAP248-286)[67]. In addition, it has been reported that the membrane-associated amyloid precursor protein (APP), which is highly expressed in macrophages and microglia, restricts HIV-1 and is counteracted by Gag-induced secretase-dependent cleavage of APP to toxic Aβ isoforms[98]. Thus, HIV-1 may promote both the formation of amyloidogenic peptides, as well as the formation of harmful amyloids by various mechanisms. In vitro Env-derived EF-C fibrils form amyloids instantaneously and it has been suggested that HIV-1 may generate its own attachment enhancers[38,67]. We found that HIV-1 particles themselves did not accelerate the formation of Aβ amyloids and it remains to be clarified whether the levels of Env-derived amyloidogenic fragments achieved in vivo are sufficient for achieving these effects. While the number of Env Spikes on virions is low, significant quantities of Env are shed from infected cells and contribute to inflammation[99]. HIV-1 gp120-derived amyloid fibrils have recently been detected in various body fluids, including CSF of AIDS patients[44], and an earlier study has shown that infusion of gp120 induced the accumulation of amyloid plaques in mice[100]. The physiological relevance of the Env-derived amyloidogenic peptides and the exact mechanism underlying the ability of EF-C fibrils to cross-seed the formation of other amyloids remains to be determined. However, studies to clarify whether the levels of Env-derived peptides in the brain or elsewhere are sufficient to boost amyloid formation by endogenous human peptides or proteins seem highly warranted.

To approximate physiological conditions, we used several HIV-1 strains and primary human cell types. Nonetheless, our cell culture analyses do not fully recapitulate the complex interactions and slowly progressing accumulation of amyloids in Alzheimer's and Parkinson's disease over several years in HIV-infected individuals. One important question is whether diseases-associated concentrations of Aβ and α-synuclein in vivo are sufficient to enhance HIV-1 infection. We used concentrations ranging from 0.4 to 50 µg/mL (0.0277 to 3.46 µM) during virion treatment corresponding to 0.08 to 10 µg/mL in the cell cultures. Concentrations of up to 2 µM of Aβ oligomers have been detected in parietal lobe human brain tissue isolated from Alzheimer's disease patients[101]. However, this number represents an average of the

whole tissue and local amyloid concentrations are certainly much higher. For example, concentrations up to 100 mg/mL of fibrillar Aβ were detected in senile plaques, while concentrations of ~4–400 µg/mL may be present around synapses[102]. α-synuclein is abundant in the human brain and is concentrated in presynaptic terminals, representing up to 1% of the total cytosolic protein content[103]. While α-synuclein is produced intracellularly, it is also released, and accumulating evidence suggests a relevant role of extracellular α-synuclein in pathological conditions[104]. Our results show that α-synuclein released from cells oligomerizes and supports HIV-1 infection (Fig. 7). Notably, it has been reported that α-synuclein is upregulated during an immune response[105], especially in the substantia nigra of HIV-infected individuals[106] and the accumulation of amyloids may result from impaired lysosome clearance triggered by Vpr[107]. Similarly, the accumulation of Aβ and the presence of amyloid plaques are common features in HIV-associated neurological disease[108,109]. In addition, impaired clearance of Aβ amyloids was shown to correlate with increased levels of HIV-1 RNA and neuroinflammation in the brain[110]. Thus, current evidence suggests that the levels of α-synuclein and Aβ amyloids achieved in vivo may be sufficient for the enhancement of virus infection.

Another key question is which types of amyloids are most relevant for the enhancement of HIV-1 infection in vivo and how these processes are linked to harmful inflammation. Our results show that α-synuclein is more potent in enhancing HIV-1 infection than Aβ fibrils in cell culture. Thus, it will be interesting to further examine whether the effects of HIV-1 infection on neurological complications are more severe in Parkinson's than in Alzheimer's disease. Our preliminary analyses showed that brain extracts from three individuals with Alzheimer's and/or LBD were not more active in enhancing HIV-1 infection than those from three individuals without neurological complications. The overall quantity of Thioflavin T-stainable material, rather than disease status, correlated with HIV-1 infection enhancement. This may be explained by the modest enhancing activity of Aβ fibrils and the presence of numerous additional aggregated factors in vivo[111]. In addition to Aβ and α-synuclein, tau, TDP-43, and fragments of TMEM106B are also commonly present in amyloid deposits in various neurodegenerative disease[112,113]. Interestingly, it has recently been shown that medin, a 50 amino acids fragment of the protein MFG-E8, interacts with Aβ to promote its aggregation and that both form

heterologous fibrils[114]. Similarly, we show that the HIV-1 Env-derived EF-C peptide co-aggregates with both Aβ and α-synuclein (Fig. 6). Thus, further studies on the interplay and effects of different types of cellular and cross-seeding of viral amyloidogenic peptides and cellular proteins are of interest.

Our finding that amyloid Aβ and especially α-synuclein boost HIV-1 attachment and infection of primary human cells, including microglia − the major viral targets in the brain − provides interesting perspectives for further investigation and development, particularly regarding neuroinflammation. It has been reported that several viral proteins (Tat, Vpr, gp120, and Nef) contribute to neuroinflammation[115]. In addition, impaired clearance of Aβ amyloid has been shown to correlate with increased levels of HIV-1 RNA and neuroinflammation in the brain[110]. Our results indicate that people living with HIV-1 may particularly benefit from anti-amyloid drugs. In addition, it will be interesting to clarify whether agents preventing the interaction of α-synuclein and other amyloidogenic agents with extracellular vesicles[116] may also abolish the interaction with viral particles and thereby mitigating the enhancing effect of amyloids on HIV-1 infection. Further studies are needed to clarify the mechanisms and therapeutic potential of the complex interactions between amyloids and viruses in animal models, infected individuals, and organoid models. These insights may ultimately help to attenuate the development of neurological disease and preserve cognitive function in people living with HIV.

## Methods

### Ethics statement
The use of human PBMCs was approved by the Ethics Committee of the Ulm University Medical Center (Approval 93/21-FSt/TR). All donors were anonymized prior to the experiments and were randomly selected from a pool of healthy donors. Informed written consent was obtained, and no compensation was provided. Sex and/or gender were not considered for the study design and were determined based on self-report. Analyses of brain samples received ethical approval by the Ulm University Ethics Committee (Ulm/Germany; Decision-No. 342/14) and by the UZ Leuven ethical committee (Leuven/Belgium; Decision-No. S-59295). In accordance with the Declaration of Helsinki, informed consent for autopsy and scientific use of autopsy tissue with clinical information was granted. All methods have been performed in accordance with the relevant guidelines and regulations.

### Peptide synthesis and fibril formation
Lyophilized peptides and proteins were commercially purchased: Aβ 40 (KE Biochem), Aβ 42 (KE Biochem, Celtek Peptides) and α-synuclein (rPeptide). The lyophilized powders were resuspended in PBS (Thermo Fisher) at concentrations of 2 or 5 mg/mL for Aβ 40, Aβ 42, and α-synuclein. Fibril formation was induced by agitation at 1500 × rpm for 5–10 days at 37 °C using an Eppendorf Thermomixer. EF-C (PTD-A) peptide was produced by standard Fmoc solid-phase synthesis, purified by preparative reverse-phase high-performance liquid chromatography (RP-HPLC), and analyzed by HPLC and mass spectrometry (MS) at the Core Facility for Functional Peptidomics, Ulm. The lyophilized powder was solubilized in dimethyl sulfoxide (Merck) at a concentration of 10 mg/mL. Fibrils formed instantly upon a 10-fold dilution of the peptide stock in PBS, as previously described[37]. EF-C seeds were generated by the sonication of fibrils for 60 sec, amplitude 50%, with pulses 1 sec on, 1 sec off. Fibril formation was verified by Thioflavin T binding assay and electron microscopy.

### ATTO NHS ester labeling of fibrils
ATTO labeling was performed as previously reported[117]. In brief, 1 mg of ATTO NHS dye was mixed with the peptide or protein stock solution and incubated for 1 h at room temperature (RT). Labeled monomer solutions were prepared by mixing 1 μL of ATTO 647 N or ATTO 495-labeled monomers with 99 μL unlabeled monomers. Cross-seeded fibrils were formed by mixing 10% (v/v) 1% labeled and sonicated EF-C fibrils with 90% (v/v) 1% labeled for Aβ 40, or α-synuclein monomers. Fibril formation was initiated by agitation in the dark at 1500 × rpm for 5–10 days at 37 °C using an Eppendorf Thermomixer. To remove unbound dye, labeled fibrils were washed by centrifugation at 20,817 × g for 15 min and resuspended in an equal amount of PBS. Fluorescence microscopy was performed using a Leica Dmi8 confocal microscope with LasX 3.7.6 software.

### ProteoStat, amytracker, and thioflavin T labeling
Labeling with ProteoStat was performed according to the ProteoStat Amyloid Plaque Detection Kit (Enzo LifeSciences). Fibrils were diluted to a final concentration of 200 μg/mL in the ProteoStat staining solution and incubated in the dark for 15 min at RT. Fibrils were stained with the Amytracker 540 dye (Ebba Biotech) according to the manufacturer's instructions by incubating the fibril solution (1 mg/mL) with 1:500 diluted Amytracker dye for 30 min at 37°C. For Thioflavin T binding, a 2.5 mM stock solution of Thioflavin T (Sigma-Aldrich) in PBS was prepared and sterile-filtered. 10 μL of fibril solution (1 mg/mL) was mixed with 1 μL of the Thioflavin T stock solution, 89 μL PBS was added and samples were incubated in the dark for 10 min at RT. Fluorescence intensity scans were performed by excitation at 450 nm and an emission of 450–650 nm for spectral scanning, or 482 nm for single wavelength analysis. To monitor the fibril formation kinetics, samples (1 mg/mL) were incubated with 25 μM Thioflavin T. The mix was pipetted into a Corning 3575 plate and two glass beads with a diameter of 1–2 mm were added to each well. For the cross-seeding experiments, samples were mixed with the indicated amounts of seeds, or sucrose-purified HIV-1. The plate was sealed and incubated at 37 °C, under continuous orbital shaking (1 mm) at a frequency of 800 cycles per minute (cpm). Fluorescence was measured with an excitation wavelength of 450 nm and an emission endpoint of 490 nm. Measurements were performed using a Synergy H1 hybrid multi-mode reader (BioTek) with Gen5 3.08 software.

### Nanoparticle tracking analysis
To measure the concentration and surface charge (zeta potential), samples were diluted in 1 mL of double-distilled water (ddH$_2$O) and measured three times using a ZetaView TWIN (Particle Metrix) with Zeta view 8.05 software. The zeta potential of fibrils was calculated based on the electrophoretic mobility of the samples.

### Transmission electron microscopy
To visualize fibrils by electron microscopy, the samples were diluted to 1 mg/mL in PBS. 5 μL of the samples were incubated for 5 min on carbon-coated formvar film, glow-discharged 300-mesh copper grids. Afterwards, washed three times with a series of three drops of ddH$_2$O and three staining steps with 2% uranyl acetate in H$_2$O were performed. The excess stain was removed using filter paper. Samples were air-dried for 1–2 h and visualized using a Jeol 1400 transmission electron microscope (Jeol) operated at 120 kV.

### Cell lines and culture
Human Embryonic Kidney (HEK) 293 T, H4 neuroglioma (H4), Microglia Clone 3 (HMC-3), Vero E6, and ELVIS cells were purchased from American Type Culture Collection (ATCC). CEM-M7, U373-MAGI, and TZM-bl cells were obtained from the NIH AIDS Reagent Program. HEK 293 T, U373-MAGI, ELVIS, and TZM-bl cells were cultured in Dulbecco's Modified Eagle Medium (DMEM, Gibco) supplemented with 10% (v/v) heat-inactivated fetal bovine serum (FBS, Gibco), 2 mM L-glutamine, 100 μg/mL streptomycin, and 100 U/mL penicillin (all PANBiotech). H4 cells were cultivated in DMEM supplemented with 10% (v/v) heat-inactivated FBS, 4.5 g/L D-glucose, and L-glutamine (Bio & SELL). HMC-

3 cells were cultivated in Eagle's Minimum Essential Medium (EMEM, ATCC) supplemented with 10% (v/v) heat-inactivated FBS. Vero E6 cells were cultivated in DMEM supplemented with 10% (v/v) heat-inactivated FBS, 100 U/mL penicillin, 100 μg/mL streptomycin, 2 mM L-glutamine, 1 mM sodium pyruvate (Gibco) + 1× non-essential amino acids (Gibco). CEM-M7 cells were cultivated in Roswell Park Memorial Institute (RPMI) 1640 medium (Gibco) supplemented with 10% (v/v) heat-inactivated FBS, 2 mM L-glutamine, 100 mg/mL streptomycin, and 100 U/mL penicillin. All cells were cultured at 37 °C, 90% humidity, and 5% $CO_2$. Adherent cell lines were passaged with 0.05% Trypsin/EDTA (PANBiotech) and used for experiments when they reached 70–80% confluence.

## Isolation of primary blood cells

Buffy coats from healthy donors were obtained from the DRK blood bank (Ulm). To obtain CD4$^+$ T cells, the buffy coats were diluted 1:2 with PBS and incubated with 50 μL/mL CD4$^+$ T cells enrichment cocktail (RosetteSep) for 20 min. The mixture was further diluted to 1:3 with PBS, layered on human Pancoll (PAN-Biotech), and centrifuged at 1200 × g for 30 min with the brakes off. The white interface layer containing leukocytes and platelets was collected and washed with PBS. Red blood cells were lysed with ammonium-chloride-potassium lysing buffer (TheraPEAK) for 5 min. After washing with PBS, 1 × 10$^6$ cells/mL were cultured in RPMI 1640 medium supplemented with 10% (v/v) heat-inactivated FBS, 2 mM L-gluta-mine, 100 μg/mL streptomycin, 100 U/mL penicillin, 10 ng/mL IL-2 (Miltenyi Biotec), and anti-CD3/CD28 Dynabeads (bead-to-cell ratio of 1:1, Thermo Fisher Scientific) for three days. PBMCs were isolated from buffy coats using human Pancoll (PAN-Biotech). To generate human monocyte-derived macrophages (MDMs), 1 × 10$^6$ isolated PBMCs/mL were cultured in 48-well tissue culture plates (Sarstedt) pre-coated with poly-L-lysine (Sigma-Aldrich) in DMEM supplemented with 10% (v/v) FBS, 100 μg/mL streptomycin, 100 U/mL penicillin, 10% (v/v) human serum (PeproTech), and 15 ng/mL M-CSF (PeproTech). To generate human monocyte-derived microglia (MMG), 1 × 10$^6$ isolated cells/mL were cultured in 48-well tissue culture plates pre-coated with poly-L-lysine, using RPMI 1640 supplemented with 1% (v/v) FBS, 100 μg/mL streptomycin, 100 U/mL penicillin, 100 ng/mL IL-34 (PeproTech), 100 ng/mL MCP-1 (Pepro-Tech), 10 ng/mL M-CSF, 10 ng/mL GM-CSF (PeproTech), and 10 ng/mL β-nerve growth factor (PeproTech). Three days later, non-adherent cells (primarily T-lymphocytes) were removed by washing with the respective cell medium. To induce the differentiation of MDMs and MMG, cells were cultured in the respective supplemented FBS-free cell medium for up to 10 or 14 days.

## Verification of macrophage and microglia markers

To analyze the expression of markers on macrophage and microglia cells by flow cytometry, cells were detached using Versene (Gibco) and washed with FACS buffer (PBS with 1% (v/v) FBS). For membrane markers, cells were stained with FITC-anti-human P2RY12 Ab (BioLegend, Cat: 392107 and isotype Cat: 400209), FITC-anti-human CD45 Ab (BioLegend Cat: 304005 and isotype Cat: 400107), Alexa Fluor® 647 anti-human IBA1 Ab (Abcam, Ab225261 and isotype Ab199093), Brilliant Violet 605™ anti-human CD4 Ab (BioLegend Cat: 317438 and isotype Cat: 400350), APC anti-human CCR5 Ab (BD Pharmingen Cat: 550856, isotype Cat: 555576), or PE anti-human CXCR4 Ab (BD Pharmingen Cat: 551510 and isotype Cat: 554689), for 1 h in a wet chamber at RT. Then, cells were washed three times in FACS buffer and fixed in 2% (v/v) paraformaldehyde (PFA). For intracellular markers, cells were permeabilized in 0.2% Triton-X-100 (Sigma-Aldrich) in PBS for 10 min at RT before the addition of antibodies. Samples were acquired using a CytoFLEX flow cytometer equipped with CytExpert 2.3 software. Acquired data were analyzed using Flowjo 10.9.0 software. To analyze the expression of markers on macrophage and microglia cells by

confocal microscopy, cells were blocked in 1% (v/v) bovine serum albumin (BSA) (Sigma-Aldrich) in TBS-T for 30 min at RT, followed by antibody staining as indicated above. Cell nuclei were stained with Hoechst for 30 min at RT. Then, the cells were washed three times in PBS and fixed in 4% (v/v) PFA. For intracellular markers, cells were permeabilized in 0.2% Triton-X-100 in PBS for 10 min at RT before the addition of the antibodies.

## Virus stocks

Infectious molecular clones of HIV-1 Bal and NL4-3 (X4) strains were obtained from NIH AIDS Reagent Program. HIV-1 AD8 was provided by Kathleen Collins (University of Michigan, Michigan, USA), and HIV-1 JRCSF by Beatrice Hahn (University of Pennsylvania School of Medicine, Philadelphia, USA), while HIV-1 NL4-3 R5 and HIV-1 NL4-3 R5-IRES-G-Luc were generated in-house. To generate virus stocks, 9 × 10$^5$ HEK293T cells were seeded 24 h before transfection in 6-well tissue culture plates (Sarstedt). Cells were transfected with 2.5 μg of plasmid DNA per well using TransIT®-LT1 (Mirus) following the manufacturer's instructions. Virus stocks were harvested 48 h post-transfection and stored at -80°C. For the fibril-forming experiment, HIV-1 was purified using 20% sucrose by layering 4 parts of freshly produced virus onto 1 part sucrose, followed by centrifugation for 3 h, 20,817 × g, at 4 °C. The virus pellet was resuspended in PBS, particle concentration was measured by nanoparticle tracking, adjusted to 3.5 × 10$^9$, and fixed in 4% PFA.

## HIV-1 infection assays

To evaluate the HIV-1 infection in the reporter cell line TZM-bl[51], 1 × 10$^4$ cells were seeded in 96-well tissue culture plates (Starstedt). The next day, HIV-1 was pretreated with different compound dilutions at a ratio of 1:1, for 3–5 min at 37 °C. Then, 20 μL of the mix was added to 80 μL medium in triplicates. Fibrils alone in the absence of HIV were also tested to ensure that they did not independently activate the HIV LTRs in the TZM-bl reporter cells. In addition, heat-treated amyloid fibrils (100°C for 1 h at 300 rpm) and HIV-1 antiretroviral drugs AMD3100 (200 nM) and maraviroc (500 nM) targeting the viral entry step were used as controls. Three days post-infection, the medium was discarded, and 40 μL of 1:4 diluted GalScreen® substrate (Applied Biosystems) in PBS containing 0.214% (v/v) Triton X-100 was added to the cells. After 45 min of incubation in the dark at RT, 35 μL were transferred to Nunc™ MicroWell™ 96-well plate (Thermo Fisher Scientific) and β-galactosidase activities were quantified as relative light units per second (RLU/s) using Orion II microplate luminometer (Tiertek-Berthold) with the Simplicity 4.20 software. Values were corrected for the background signal derived from the uninfected cells. 1 × 10$^4$ CEM-M7 cells were seeded and infected as described above for TZM-bl cells. Three days post-infection, cells were washed with PBS and fixed in 4% PFA. In all the experiments, GFP+ gates were set based on the uninfected cells treated in parallel. Samples were acquired on a CytoFLEX flow cytometer equipped with CytExpert 2.3 software. Acquired data were analyzed using CytExpert 2.3. U373-MAGI and HMC-3 cells were seeded at a density of 1 × 10$^4$ cells/well in 96-well tissue culture plates one day before infection, and infected as described above for TZM-bl cells. Two hours post-infection, the inoculum was washed off and fresh medium was added. As a Gaussia luciferase reporter virus was used, supernatants were collected immediately to represent the washing control, further collected every three days, and frozen at -80 °C. For the infection of primary cells, 1 × 10$^5$ CD4$^+$ T cells were seeded in 96-U-well plates (Sarstedt). MDMs and MMG cells differentiated in 48-well plates as described above, were infected with a ratio of 1:4 (v/v) ratio of HIV-1 pretreated with different compound dilutions (1:1), for 3–5 min at 37 °C. To determine the infectivity of virions produced in primary blood cells, ~50% (v/v) of supernatants of primary CD4$^+$ T cells and ~30% (v/v) of supernatants of MDMs and MMG cultures at the indicated time points were collected and frozen at -80 °C. TZM-bl cells were

seeded in 96-well plates (Sarstedt) at a density of $1 \times 10^4$ cells in 100 μL per well and infected with 25 μL of the supernatants collected from primary CD4$^+$ T cells, MDMs, or MMG. Three days post-infection, viral infectivity was determined using the β-galactosidase screen kit as described above.

## HIV-1 infection in trans

To examine HIV-1 in trans-infection, $5 \times 10^3$ HMC-3 cells were seeded one day prior to infection in 96-well plates (Sarstedt). The next day, 20 μL of a 1:1 fibril-virus mix was added to 80 μL of medium. After 4 h of incubation, the cells were washed three times with PBS, and $5 \times 10^3$ TZM-bl cells were added in 200 μL of medium. As a Gaussia luciferase reporter virus was used, supernatants were collected prior to the addition of TZM-bl cells, representing the washing control, further collected every three days, and stored at −80 °C.

## Gaussia Luciferase assay

To measure the infection of Gaussia luciferase reporter HIV-1, coelenterazine substrate (PJK GmbH) was prepared by dissolving 1 mg in acidified methanol (1 drop of concentrated HCl to 10 mL of methanol), and stored at −80 °C. For the measurements, 100 μL of the substrate diluted 1:120 in PBS was added to 20 μL of supernatant, and activity was quantified as RLU/s using the Orion II microplate luminometer (Tiertek-Berthold) with Simplicity 4.20 software. For the bioluminescence complementation assay with ASYN-hGLuc1 (S1) and ASYN-hGLuc2 (S2) constructs, cells were transfected as described above, and conditioned media was collected 48 h post-transfection. Luciferase activity from protein complementation was measured using 100 μL of conditioned media in an automated plate reader (Victor X3 microplate reader, PerkinElmer) at 480 nm, using a signal integration time of 1 sec after the injection of 100 μL coelenterazine (1 mg/mL, PJK GmbH).

## ZIKV infection

6000 Vero E6 cells were seeded the day before infection. Fibrils at indicated concentrations were incubated with ZIKV MR766 at an MOI of 0.3 for 5 min at 37 °C. The mixture was either directly added to cells or centrifuged for 10 min at 3220 × g, separated into supernatant and resuspended pellet, and then added to cells. Two days later, infection rates were determined using a cell-based ZIKV immunodetection assay. First, cells were washed with PBS and fixed with 4% PFA for 20 min at RT. Then cell permeabilization was performed with cold methanol for 5 min at 4 °C, followed by washing with PBS. Then, cells were incubated with mouse anti-flavivirus antibody 4G2 (Absolute Antibody Cat: Ab00230-2.0) in antibody buffer (0.3% (v/v) Tween 20, 10% (v/v) FBS in PBS) for 1 h at 37 °C, washed three times with washing buffer (0.3% (v/v) Tween 20 in PBS), and incubated with horseradish peroxidase (HRP)-coupled anti-mouse antibody (Thermo Fisher Scientific Cat: A16066) for 1 h at 37 °C. After four washing steps with washing buffer, 3,3′,5,5′-Tetramethylbenzidine (TMB) substrate (Medac) was added. Following an incubation of 5 min at RT, the reaction was stopped with 0.5 M sulfuric acid, and absorbance was measured at 450 nm, with and baseline correction at 650 nm using a VersaMax ELISA microplate reader (Molecular Devices) with SoftMax 7.0.3 software.

## HSV-2 infection

5000 ELVIS cells[66] were seeded the day before infection. Fibrils at the indicated concentrations were incubated with HSV-2-GFP 333 at a MOI of 0.1 for 5 min at 37 °C. The mixture was either directly added to cells or centrifuged for 10 min at 3220 × g, separated into supernatant and resuspended pellet, and then added to cells. One day post-infection, the rates of infection were measured using the β-galactosidase assay as described above.

## Virion fusion

HIV-1 BlaM-Vpr particles were generated by cotransfection of HEK293T cells with HIV-1 AD8 proviral DNA, pCMV-BlaM-Vpr, and pAdVAntage vectors, as previously described[62]. HIV-1 virion fusion was assessed based on the incorporation of a β-lactamase Vpr (BlaM-Vpr) fusions into the virions and its subsequent transfer into a target cell where CCF2, the fluorescent dye substrate of β-lactamase is cleaved. Cleavage upon entry of the BlaM-Vpr-containing viral capsids into the cell triggers a fluorescence shift of CCF2 from green (520 nm) to blue (447 nm), after excitation at 405 nm. To quantify HIV-1 fusion in PBMC-derived macrophages and microglia, $2.5 \times 10^5$ freshly isolated PBMCs were seeded in 8-well Ibidi slides (Ibidi) pre-coated with poly-L-lysine (Sigma-Aldrich). On day 10 or 14 of differentiation, cells were infected with 100 μL of 1:1 fibril-virus mix in 100 μL medium for 6 h at 37 °C. Then, the cells were washed three times with PBS, and loaded with CCF2 dye and the reagents from the Beta-lactamase Loading Solutions Kit (Invitrogen) (0.3 μL CCF dye, 1.2 μL solution B, and 1.5 μL solution D in 150 μL $CO_2$-independent medium/sample) and incubated overnight at RT in a wet chamber. On the next day, cells were washed three times with PBS, fixed in 4% PFA, and imaged by fluorescence microscopy using a Leica Dmi8 confocal microscope (Leica) with LasX 3.7.6 software. The mean fluorescence of the green and blue channels was quantified using Fiji ImageJ 9.1 software.

## Virus–fibril interaction

Fibrils were stained with the Amytracker 540 dye (Ebba Biotech) according to the manufacturer's instructions. HIV-1 particles were generated using a CFP labeled HIV-1 gag construct and YU-2 strain HIV-1 envelope as previously described. To detect the interaction of the fibril-virus mix with cells, $3 \times 10^4$ TZM-bl cells were seeded in 8-well slides (Ibidi) in 160 μl medium one day before. Stained fibrils at a concentration of 50 μg/mL on virus were pre-incubated with HIV particles for 3–5 min at 37 °C, and 40 μL of the mixture were transferred to an Ibidi μ-slide chamber (Ibidi) for visualization. The mixture was incubated with the cells for 2 h to allow binding, washed three times with PBS, and fixed in 4% PFA. For cell staining, the CellTrace Violet Cell Proliferation Kit was used, and cellular actin was stained with the ATTO-647-phalloidin dye (ATTO-Tec).

For the experiments with Murine Leukemia Virus (MLV), ProteoStat-stained fibrils were pre-incubated with diluted MLV GAG YFP virus-like particles (VLPs) for 3–5 min at 37 °C, and 30 μL of the mixture were transferred to an Ibidi μ-slide chamber (Ibidi) for visualization. To detect the interaction of the fibril-virus mix with cells, $3 \times 10^4$ TZM-bl cells were seeded in 8-well slides (Ibidi) one day before the addition of amyloids, viral particles, or mixtures thereof. For cell staining, the Cell-Trace Violet Cell Proliferation Kit (Invitrogen) was used. For both experiments, fluorescence microscopy was performed using an LSM 710 confocal microscope (Zeiss) with Zen Black Studio 2010 software.

## Fibril-virus pull-down assay

To determine the binding between fibrils and virus, HIV-1 was incubated with different compound dilutions for 3–5 min at 37 °C, then the mixture was pelleted by centrifugation at 3220 × g for 10 min. A total of 20 μL of the supernatant was carefully collected, and the pellet was resuspended in 20 μL of medium, and the supernatant and resuspended pellet were added to TZM-bl cells. Three days post-infection, β-galactosidase assay described above was performed.

To quantify HIV-1 p24 antigen levels, an in-house ELISA was used. Briefly, 96-well microplates (Nunc Immuno Plate, MaxiSorp Surface) were coated with 500 ng/mL anti-HIV-1 p24 antibody (Abcam, Cat: 43037) and incubated overnight at RT in a wet chamber. The following day, plates were washed three times with PBS-T (PBS containing 0.05% (v/v) Tween 20) and then blocked with blocking solution (PBS containing 10% (v/v) FCS) for 2 h at 37 °C. After washing, 100 μL of serially diluted HIV-1 p24 protein standards (Abcam, ab43037) and samples

(lysed with 1% (v/v) Triton-X-100) were added to the wells. Plates were incubated overnight at RT in a wet chamber. Next day, unbound capsid proteins were removed by washing, and 100 μL/well of polyclonal rabbit antiserum against p24 (Eurogentec) (diluted in PBS-T with 10% (v/v) FCS) was added for 1 h at 37 °C. After washing, 100 μL of HRP-coupled anti-rabbit antibody (Dianova, Cat: 111-035-008) was applied for 1 h at 37 °C. Finally, plates were washed, and 100 μL of TMB substrate was added. After 20 min of shaking at 450 rpm at RT, the reaction was stopped with 50 μL of 0.5 M sulfuric acid. The optical density, correlating with p24 capsid antigen levels, was measured at 450 nm and baseline corrected at 650 nm using a VersaMax ELISA microplate reader and SoftMax 7.0.3 software, and quantitative results were calculated based on a standard curve.

### Generation of α-synuclein supernatants

H4 cells were cultivated as described above and seeded in 6-well plates (Sarstedt) 48 h prior to transfection. The ASYN-hGLuc1 (S1) and ASYN-hGLuc2 (S2) fusion constructs (wildtype and mutants) were generated as previously described[118]. The Gaussia luciferase is split into two parts and each part is fused to either wild-type α-synuclein or one of the mutant constructs (A30P, E46K, A53T). Upon α-synuclein oligomerization, the reporter parts form a functional reconstituted bioluminescent active luciferase. H4 cells were transfected with equal amounts of split ASYN constructs, or a pcDNA empty vector as a control using PolyFect Transfection Reagent (Qiagen) according to the manufacturer's instructions. Opti-MEM reduced serum medium supplemented with L-glutamine and HEPES (Gibco) was used as the growth medium. Cells were further cultured for 48 h before media was collected for HIV-1 infection assays. For the depletion of α-synuclein in the supernatant of wild-type ASYN transfected cells, 150 μL protein G Mag Sepharose Xtra beads (GE Healthcare) were supplemented with 1 mL Opti-MEM reduced serum medium supplemented with L-glutamine and HEPES and rotated at 4 °C for 10 min. The beads were centrifuged at 20,817 × g for 1 min and placed in a magnet (Invitrogen). The supernatant was discarded and the washing step was repeated a total of three times. In a pre-clearing step, 50 μL of washed beads were incubated with 700 μL of conditioned media for 1 h at 4 °C on a rotator (Stuart). The beads were centrifuged at 20,817 × g for 10 min and placed in a magnet to transfer the supernatant to a new tube. 5 μg SYN-1 antibody (BD Bioscience, Cat: 610787) was added to the conditioned medium and rotated for at least 3 h at 4 °C. The remaining 100 μL of washed G beads were then added and rotated overnight at 4 °C. The next day, the sample was centrifuged at 20,817 × g for 10 min and the supernatant was used as α-synuclein depleted media.

### Processing of brain lysates

Human brain lysates were obtained from post-mortem tissue of freshly frozen human occipital (Brodmann areas 17–19) and temporal cortex (Brodmann areas 35 and 36). One donor was diagnosed with Alzheimer's disease, one with LBD, and one with both, while the remaining three donors represented age-matched control individuals without neurological disorders (Supplementary Table 1). Samples were homogenized by sonication, and the insoluble fraction was separated by centrifugation at 14,000 × g, for 2 h at RT, and discarded. The supernatants containing the soluble fraction were retained and normalized by protein concentration. Then, TZM-bl cells seeded one day prior at a density of $1 \times 10^4$ cells/well, were infected with a 1:1 mix of titrated brain extract and HIV-1 NL4-3 R5 tropic strain. The inoculum was washed after 3 h, two days post-infection, the β-galactosidase assay described above was performed.

### Statistical Analysis

Statistical analyses were performed using GraphPad PRISM 10 (GraphPad Software). P values were determined using a two-sided Student's t-test with Welch's correction and 95% confidence interval or one-way or two-way ANOVA with Dunnett's multiple comparison test. Unless otherwise stated, data are shown as the mean of at least three independent experiments ± SEM. Significant differences are indicated as: *$p < 0.05$; **$p < 0.01$; ***$p < 0.001$. Statistical parameters are specified in the figure legends.

### Reporting summary

Further information on research design is available in the Nature Portfolio Reporting Summary linked to this article.

## Data availability

All data are available in the main text or the supplemental information. Source data are provided with this paper. The image data generated in this study have been deposited in the Figshare database under accession codes https://doi.org/10.6084/m9.figshare.28053890.v1, 28053896.v1, 28053917.v1, 28053971.v1, 28053980.v1, 28053986.v1, 28053995.v1, 28053998.v1 and 28054007.v1. Source data are provided with this paper.

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

## Acknowledgements

SL is part of and acknowledges the support from the International Graduate School in Molecular Medicine Ulm (IGradU). We'd like to acknowledge the DFG project 432000323 for using the Leica DM8i microscope. Moreover, we'd like to thank and acknowledge the technical assistants of the Institute of Molecular Virology for their support. This work was supported by grants from the German Research Foundation (DFG CRC 1279 and CRC 1506). The preparation and characterization of the human tissue samples was funded by DFG TH624-6-1 (DRT), Alzheimer Forschung Initiative (#10810 (DRT), Fonds Wetenschappelijk Onderzoek Vlaanderen (FWO: G0F8516N, G065721N (DRT)), and Alzheimer Association (22-AAIIA-963171 (DRT)). The funders had no role in study design, data collection and analysis, decision to publish, or preparation of the manuscript.

## Author contributions

Conceptualization: F.K., J.M., L.R.O.; Methodology: L.R.O., F.A., M.G.M.; Investigation: L.R.O., S.L., M.G.M., F.A., J.K., C.S.; Resources: D.R.T., A.R.U., P.W.; Supervision: F.K., J.M., K.D.; Writing—original draft: FK; Writing—review & editing: F.K., J.M., K.D., K.M.J.S., D.R.T., L.R.O., S.L.

## Funding

## Competing interests

DRT collaborated with Novartis Pharma AG (Basel, Switzerland), Probiodrug (Halle (Saale), Germany), GE Healthcare (Amersham, UK), and Janssen Pharmaceutical Companies (Beerse, Belgium). JM and FK are inventors on patents for using peptide nanofibrils to enhance viral transduction. Remaining authors report no competing interests.
