## [Transparent Peer Review file · Nature Communications]

alpha-Synuclein fibrils enhance HIV-1 infection of human T cells, macrophages and microglia

Corresponding Author: Professor Frank Kirchhoff

Version 0:

Reviewer comments:

Reviewer #1

(Remarks to the Author)

The manuscript explores the role of amyloid fibrils in various neurological disorders, particularly their potential involvement in HIV-associated neurological disorders (HAND). While amyloid fibrils are known for their association with Alzheimer's, Parkinson's, and Huntington's diseases, they are also significant in HIV-related conditions. Olari et al. describe how amyloid aggregation may accelerate neurocognitive impairments in HIV-infected individuals, potentially exacerbating HAND. This study aims to understand how amyloid deposits influence HIV-1 infection in human cells, highlighting the roles of amyloid A β and α -synuclein in the brain. While the study is very insightful and timely, there are some technical and conceptual concerns that need to be addressed.

1. The manuscript claims the use of statistical analyses, specifically the Student's t-test, but this is not evident in the figures. Figures 1B, 1D, 1E, 2A, 2B, 2C, 3A, 3B, 5C, 6E, and 7 lack statistical annotations, yet the results are interpreted. This needs to be rectified.
2. Figure 2 shows that A β 40 did not affect HIV-1 infection, while α -synuclein fibrils increased infection only at the highest dose. Given that astrocytes experience abortive infection, did the authors check for viral RNA integration into the genome to confirm infection?
3. The manuscript does not mention the source of monomeric or oligomeric α -synuclein used in the study.
4. Heat-inactivated A β 40 and α -synuclein should be included as negative controls in the study.
5. The authors should define the abbreviation EF-C and clarify the rationale for its use. Is it a positive control?.
6. Elaborate on the role of A β and α -synuclein in HIV infection to set up a premise for the study.
7. The introduction effectively outlines the involvement of amyloids in various diseases but could benefit from a more detailed discussion on their dual roles in normal physiology and disease for a balanced view.
8. How do the amyloid interactions described in this study compare with other known interactions in similar or different diseases?
9. How specific are the interactions between α -synuclein or A β 40 fibrils and HIV-1? Could these interactions also affect other viral pathogens similarly?
10. Authors need to discuss What molecular mechanisms underlie the increased HIV-1 infection rates in the presence of amyloid fibrils? What concentrations of amyloid fibrils were used, and how do these compare to physiological levels in neurodegenerative diseases?
11. Did the authors perform a dose-response study for each type of fibril to better understand the relationship between fibril concentration and HIV-1 infection enhancement?
12. Please clarify the selection criteria for the cell lines and HIV strains used in the experiments.
13. What are the potential therapeutic implications of the findings for HIV treatment or prevention, particularly regarding neurocognitive impairments?
14. Mention the key limitations of the study and how they might affect the interpretation of the results.
15. The authors should discuss future studies to explore further the connections between amyloid fibrils and HIV infection or other neurological conditions.
16. Minor grammatical errors need correction, and some sentences should be simplified for clarity and impact.
17. Detailed patient demographics should be provided better to understand the effects of these proteins on HIV replication.
18. Clarify if the detrimental effects of A β and α -synuclein on HIV infection correlate with the severity of AD and PD.

(Remarks to the Author)

The manuscript by Olari et al investigate an intriguing question of whether amyloid fibrils associated with Alzheimer and Parkinson disease, amyloid beta and alpha synuclein, enhance HIV infection in T-cells and myeloid lineage cells. Previous evidence the Semen-derived Enhancers of Viral Infection (SEVI), which promote HIV-1 infection, form amyloid fibrils and suggest that amyloid beta and alpha synuclein amyloid fibrils may act similarly in the brain and contribute to the enhanced neuropathogenesis of HIV. The premise of this manuscript is relevant to understanding the impact of aging in people living with HIV(PWH). This is increasingly important as the proportion of PWH in the US over the age of 50 continues to increase and will be the majority of the population in the coming decade. The manuscript includes promising data that there is enhanced activation of the HIV LTR in response to amyloid exposure. However, additional controls and assays are needed to draw the conclusions stated in this manuscript that alpha synuclein fibrils enhance HIV infection. The need for these controls is further supported by the data using patient samples that does not show specificity for alpha synuclein amyloid fibrils in induction of the HIV LTR. The following critiques are offered to help substantiate the conclusions drawn in the manuscript and support this novel hypothesis.

Major Critiques.

1. A key missing control for several experiments is the effect of the amyloid fibrils (AB, alpha synuclein and EF-C) without HIV on the viral reporter cell lines and primary cells. As the main assay for "HIV infection" is the readout of HIV LTR transcription via a reporter gene, it is important to demonstrate that the amyloid fibrils themselves are not activating the HIV LTRs in the reporter cells as the LTRs contain a number of cis-binding elements that are activated by NF-kB and early response elements that can be activated by numerous signaling cascades. There is a robust literature to suggest that both beta amyloid and alpha synuclein interact with the toll like receptor and NLRP3 inflammasome pathways which would result in activation of the HIV LTRs even in the absence of viral infection. Without this control, it is difficult to specifically interpret the data from Figure 2 and the subsequent figures to be specific to HIV infection.
2. It would be valuable to show the baseline effect of HIV infection alone for all assays to assess the robustness of the infection prior to the addition of the fibrils. The data are often presented as a fold change from baseline that precludes this assessment.
3. When using primary cells to infect with HIV, a critical complementary assay for HIV infection is production of p24. This assay would be important not only for Figure 3 to demonstrate that the primary cells are producing virus, but also for Figure 2C co-culture experiments to demonstrate that virus is produced from the infected cells prior to co-culture with the fibrils and the second cell type. Otherwise, the observation may be attributed to the activation or the inflammasome in the infected cell type and chemokine signaling activating the LTR in the second cell type. Controls to address this possibility should be added.
4. Figure 3c suggests that the change in HIV LTR activation in microglia by co-incubation with amyloid fibrils is not related to HIV infection and exhibits the opposite kinetics to support this conclusion. The additional controls suggested may clarify this outcome and the interpretations and title of the manuscript should be modified to reflect that.
5. Use of HIV antiretroviral drugs are important controls that would be valuable in determining the contribution of HIV replication to the readouts (with appropriate controls added for the impact of these drugs on the cell types being used.)
6. Caution should be exercised with the interpretations of the assay that only readout activation of the HIV LTR. As an example, in figure 4, the conclusion is drawn in lines 185-187 "These results support that amyloid alpha synuclein enhance HIV-1 replication in macrophages and microglia cells by enhancing virion attachment and fusion with these target cells." as this is not demonstrated by simply observing the activation of the LTR in macrophage/microglia by expression of the reporter gene. Figure 5 tries to get at this; however, the use of murine leukemia virus to show interaction between the amyloid fibrils and this virus cannot be transferred to indicate that HIV interacts with the fibrils. The experiment would need to include a condition with HIV colocalizing with the amyloid on the surface of the infected cells.
7. Due to the indirect nature of the assay in Figure 5 C, the conclusion drawn in lines 206-208, " Altogether, these results show that α -synuclein fibrils bind HIV-1 and promote virus infection similarly to EF-C amyloids, while A β fibrils are less effective." are not substantiated by the data which support MLV interaction with the fibrils (5a/b) and show LTR activation kinetics of the fibrils mixed with HIV and exposed to reporter cells without the HIV on its own. The use of an HIV antiretroviral from the entry inhibitor class would aid in this conclusion.
8. For several of the figures, it is unclear how the authors are determining a significant outcome as statistical methods are not outlined in the figure legends and significant findings are not indicated in the figure, particularly in Figures 2, 3, 5, and 6).
9. For Figure 7, important controls to show contribution of alpha synuclein would include cells that do not express alpha synuclein or the use of an alpha synuclein neutralizing antibody controls to show alpha synuclein specificity.
10. The human data do not support the hypothesis of the paper. The data are interpreted as not showing a difference between the extracts from patients with AD, LBD or control. However, as these data do not support the data using laboratory derived fibrils, this should be acknowledged and discussed instead of spun in a way to try to validate the findings from the laboratory derived treatments over the human derived materials.

11. For Figure 8, controls to rule out the effects of non amyloid or non viral effects of human tissue on activation of the LTR are needed given the abundance of materials that could activate the NLRP inflammasome in the human lysates.

12. The discussion should be modified to align with more cautious interpretations given the indirect nature of the assays interpreted as HIV infection and the use of transformed cell lines used for these studies.

Minor points

Line 145-147 "These results suggest that amyloids may be more important for efficient spread of primary R5-tropic HIV-1 strains than for late-stage X4-tropic HIV-1 variants" This statement is an overinterpretation of the data that do not verify viral production. Further this statement should be for the discussion.

Line 179-180 "Transfer into the target cells upon fusion is monitored by enzymatic cleavage of CCF2 by β -lactamase" It would be helpful to explain the biologic implication of this assay as opposed to the specific substrate/enzyme. This would aid in interpretation of the data for the reader.

Line 228 and 232, monogenic and heterogenic are not the best word as these are not genes, but protein mixtures.

Reviewer #3

(Remarks to the Author)

In this study, Olari et al. report that incubating fibrils derived from alpha-synuclein or to a lesser extent Ab40 enhance HIV-1 infection. While the findings are interesting, they are not novel, the findings are undeveloped, there is a lack of detail, and many important controls are missing to support the conclusions being drawn. In addition, the underlying mechanism and biological relevance of the findings is unclear.

Main Concerns

It is well established that fibrils enhance infection and some studies of HIV and alpha-synuclein are not cited. The vast majority of this study just involves mixing virus preparations with fibrils and simply showing the final effect on infectivity, but the preparations and relevance are not characterized at all. From the methods, it seems that the virus and fibril are just mixed and added to cells. The mixed preparations need to be characterized in detail. How much virus is bound to fibrils and how much is not? How much fibril is not bound to virus?

The images in figure 5 do not address this issue and instead raise more questions. First off, they switch to MLV rather than HIV. Second, the images are poor quality as, for example, the cytoskeleton stain is completely lacking in detail and just shows a huge blob where the cell is. The individual images are not from the same fields of view in, for example, the top panels in 5a, or if they are, it is not clear that they are and whether the MLV fibril panel is a zoom of a specific region. There is no scale bar on this particular image, but from this and other images, the fibrils relative to the size of virus particles or cell seem enormous compared to the sizes shown in figure 1. This would suggest that the fibrils are being added in enormous excess. Is this physiologically relevant? How do they promote infection? Do they activate cell signalling, for example? The proposed mechanism is an assumption and based on prior literature, making it uncertain and not particularly novel.

During a natural infection, how much alpha-synuclein is made and is it secreted? How is it made during infection? Or is this proposed to be a passive consequence of infection of the brain where these products are made? In which case, what are the physiologically relevant levels of these fibrils as they are unlikely to be as abundant as shown in figure 5.

The studies of EF-C and how it enhances fibril formation have little relevance to actual infection as it is not clear that EF-C is made naturally. My understanding is that it is an artificial peptide used to promote retroviral transduction.

The use of gaussian luciferase-based fibril transfection systems is interesting but artificial and similar to the mixing experiments, is limited in terms of scope, characterization and relevance.

The trans-infection assays need to be better controlled. The authors state that the microglia cell line that is used cannot be infected, but monocyte-derived microglia (MMGs) can. But there could be a low level of infection in line used for the trans-infection assay. This assay extends for a much longer period of time and although it uses TZM-blam as recipient cells, the assay uses viral gaussian as a readout of infection, which does not tell you which cell type is infected. Throughout the paper, there is no indication of MOI or the titers of different virus stocks and strains. If the microglia line is just infected at less efficiently than other cell lines, the virus may be spreading in them slowly and infecting TZM cells at the same time during this longer experimental setup. There is no evidence for the claim of trans-infection in this setup, or of the general conclusions being made.

Antibody-mediated depletion is used in several approaches but in no case is the depletion efficiency demonstrated and control depletions or antibodies are missing. Similarly, where peptides or fibrils are added, control peptides rather than

nothing at all should be added.

The use of human brain homogenates is interesting but again, they are heterogenous in nature and poorly characterized and controlled. These preparations contain a lot of things that could influence infection. The depletion approach is again uncontrolled. There is no evidence of actual depletion or the extent to which depletion occurs to correlate with effects on infection. Control antibodies need to be used also. The effects are just correlative in their current state.

THT is a very broad fibril staining approach that does not discriminate between different types, particularly when using it to characterize these brain extracts. More specific measures of each fibril type are needed. In addition, where THT is used in experiments, does it affect fibril structure or binding to virus?

Several experiments are performed just once and measured several times to generate data points and statistics. In other cases, the number of independent repeats is unclear. For imaging in figure 4, for example, 5 different images are analysed which is not a lot, and it is unclear if this experiment was done just once.

Minor Comments

Brain-derived HIV strains may be better for several of the cell types being studied.

The authors use soluble fibrils but, in many cases, fibrils are thought to be insoluble and form plaques. This should be discussed to better understand the relevance of the choice.

Line 77: EF-C amino acids numbers in the text does not match that of the table in Figure 1a.

Line 88-82 is confusing, are the authors making a structural comparison between the fibrils?

Suppl Figure 3b: The staining for P2RY12 is not strong and is unconvincing. It seems equivalent in both macrophages and microglia, which does not align with the claim that it is only expressed by microglia or what is shown for the FACS analysis.

Version 1:

Reviewer comments:

Reviewer #1

(Remarks to the Author)

The authors revised the manuscript satisfactorily.

Reviewer #3

(Remarks to the Author)

The authors approach of undermining the reviewer's expertise and understanding is not particularly helpful (both this reviewer and others), and in some cases seems to be used to avoid addressing comments. Despite this, the revised manuscript now better explains the potential significance. However, I do think some issues remain that need to be clarified.

The authors avoid addressing comments about the quality of images of MLV and fibrils in figure 5, and simply move it to the supplemental data. The regions at the same magnification (namely 10um bars) are from separate images and the merges are confusing. For example, in Supplemental Figure 5a the image of "MLV, Cells" and "MLV, fibrils, Cells" is clearly not a merge of the "MLV, fibrils" image, as the fibrils are different in each image, as is the MLV alone image. This is unconventional and confusing, what is being presented here? The same is occurring for supplemental 5b.

The new images of HIV in figure 5b are just merges that lack detail. It looks like HIV forms huge clumps presumably on fibrils, but it is unclear how much of the orange signal is from the fibrils versus overlap with virus particles. It cannot be 100% overlap and separate images should be shown of what HIV particles look like under these conditions, as is done for MLV.

The authors argue that they did some experiments once but analyzed five cells. If the experiment itself had an issue then all five cells are erroneous, as is the interpretation. The whole experiment should be repeated at least a second time to ensure this is not the case, or discuss this in light of other findings that support the conclusion independently.

The authors argue that the depletion approach for a-synuclein must have worked because it reduced HIV infection to avoid addressing the question of verifying that the depletion worked, was specific and correlates with the effect on infection. Reduced infection could be due to any number of issues, such as the antibody itself neutralizing the virus, and it seems important to verify that the extent of a-synuclein depletion correlates at least to some extent with the effects on infection.

The trans-infection assay remains quite crude and overinterpreted, as noted by other reviewers. The transfer event could very easily be from the virus bound to fibrils on the surface and nothing to do with the cell type transferring virus. Adding fibrils alone, without cells, will also increase infection so the HCM-3 cells could simply be fibril carriers. Overall, the interpretations are speculative particularly as the authors use a singular readout of infection for two different mixed cell types, and the biological relevance is unclear. This assay doesn't add much to the report.

Reviewer #4

(Remarks to the Author)
Review

This is a highly interesting extension of previous findings (Wojtowicz et al. 2002) that amyloid-derived peptides enhance the attachment and infection by HIV-1 and other enveloped viruses including MuLV. Furthermore, Münch et al. (Dr. Kirchoff group) in a Cell paper (2007) previously showed that amyloid fibrils in semen enhanced HIV-1 infection. The current findings now extend this finding to show that a-synuclein, a presynaptic protein linked neuropathologically to Parkinson's disease, also enhances HIV-infection, possibly by the same mechanism of increasing adsorption or adherence to cell membranes, but not Abeta40. The results are not entirely unexpected, but do point to the impact of amyloid fibrils in novel settings such as Parkinsons and HIV.

In the introduction section, it is stated that "the levels of amyloid deposition correlate with viral loads" – with a citation of two review articles. This is a very important finding that would set the stage for the current manuscript. The authors should cite the original references. It was impossible for this reviewer to find the original references within the two review articles cited.

The manuscript essentially shows that a-synuclein, but not Abeta40 fibrils (but not their monomers), enhance HIV-infection of a variety of target cells including CD4 T cells, monocyte derived macrophages and monocyte-derived microglia. The primary influence of a-synuclein fibrils appears to be in enhancing the virion attachment to the membrane. The binding or capture of HIV to the plasma membrane can be achieved even without CD4 receptor, but in that case it remains bound to the cell but does not enter the cells. But if CD4 and a coreceptor are present, the virus appears to proceed and complete the infection. The fluorescence images show an increased attachment of the virus, when a-synuclein fibrils are present, to the cell surface. Vpr-beta lactamase-based fusion assay results show that the fusion is enhanced. Although the fusion is the downstream step, this is expected.

I failed to see the importance of the findings reported in Figure 6. As a technique to enhance the infection by HIV-1, the role of EF-C can be appreciated (already published and cited in the manuscript). But the question being addressed here, whether EF-C enhances the rate of aggregation of either Amyloid or the a-synuclein – appears something out of a pure technical point of view. It does not add to the mechanistic understanding of the phenomenon being reported.

The responses of the authors to the Reviewer 2 comments are all acceptable – except those for #10. In their responses to point #10 by Reviewer 2 – "The human data do not support the hypothesis of the paper" and "this should be acknowledged and discussed instead of spun in a way to try to validate the findings from the laboratory derived treatments over the human derived materials" – the authors commented as follows. Author comments are in quotes followed by my remarks.

- a. "the results using human brain samples are very preliminary".
- b. In addition, they "could not determine which types of amyloids were present"
- c. "Definitive conclusions about the roles of different types of amyloids in HAND will require large human studies"
- d. "results show that enhancement of HIV-1 infection correlates with the levels Thioflavin-stainable material in brain-derived samples"

On the one hand the authors appear eager to present information that agrees with their in vitro data. On the other hand, they know that their human results are very preliminary and have many caveats. Therefore, I agree with the Reviewer 2 on this point. The authors should acknowledge these shortcomings (a, b, c and d above) in the text and allow the reader to draw their own conclusions.

Version 2:

Reviewer comments:

Reviewer #3

(Remarks to the Author)

I find the authors approach to responding to comments extremely frustrating and off-putting. Lecturing on the quality of imaging and the reviewer's lack of understanding simply aims to hide the fact that they never provided conventional unmerged images to allow proper assessment of the data, while similar petty arguments over where 5 cells versus 5 fields of view were imaged is designed to hide the fact that the experiments were not repeated in the prior submission, which could have meant all data - whether cells or fields - was erroneous.

However, the authors have addressed my main concerns and I am glad that they are happy that I now understand their paper.

Response to the reviewer`s comments (in *italic* letters)

Reviewer #1: The manuscript explores the role of amyloid fibrils in various neurological disorders, particularly their potential involvement in HIV-associated neurological disorders (HAND). While amyloid fibrils are known for their association with Alzheimer's, Parkinson's, and Huntington's diseases, they are also significant in HIV-related conditions. Olari et al. describe how amyloid aggregation may accelerate neurocognitive impairments in HIV-infected individuals, potentially exacerbating HAND. This study aims to understand how amyloid deposits influence HIV-1 infection in human cells, highlighting the roles of amyloid A β and α -synuclein in the brain. While the study is very insightful and timely, there are some technical and conceptual concerns that need to be addressed.

We thank reviewer 1 for the positive comments and addressed all concerns, as outlined below.

1. The manuscript claims the use of statistical analyses, specifically the Student's t-test, but this is not evident in the figures. Figures 1B, 1D, 1E, 2A, 2B, 2C, 3A, 3B, 5C, 6E, and 7 lack statistical annotations, yet the results are interpreted. This needs to be rectified.

To address this, we added statistical annotations to all figures using Student's t-test or two-way ANOVA with Dunnett's Multiple Comparison where appropriate. Initially, we did not include statistics in some figures showing several curves because they were already very busy. To address this concern, while keeping the figures clear, we color-coded the curves and annotations in the revised manuscript.

2. Figure 2 shows that A β 40 did not affect HIV-1 infection, while α -synuclein fibrils increased infection only at the highest dose. Given that astrocytes experience abortive infection, did the authors check for viral RNA integration into the genome to confirm infection?

There seem to be some misunderstandings. A β 40 fibrils enhanced infection of TZM-bl cells (revised Fig. 2a, new supplementary Fig. 1a) and in HMC/TZM-bl cocultures (Fig. 2d) by several HIV-1 strains about 3-4-fold at the highest dose of 50 μ g/ml. In comparison, α -synuclein increased HIV-1 infection in a dose-dependent manner starting at 2 μ g/ml and up to two orders of magnitude at the highest dose. We focused on microglia cell, macrophages and T cells and did not examine astrocytes since they are not productively infected by HIV-1. As mentioned in the manuscript (lines 67-69; 170-172, 341-343) microglia cells are the main target cells for HIV-1 replication in the brain and essential for infection of brain organoids (e.g. Kong et al., 2024 and own unpublished data).

3. The manuscript does not mention the source of monomeric or oligomeric α -synuclein used in the study.

Monomeric α -synuclein was purchased from rPeptide (line 438) and amyloids were generated as specified in the Material and Methods (lines 436-448).

4. Heat-inactivated A β 40 and α -synuclein should be included as negative controls in the study.

We thank the reviewer for this suggestion and now show that heat-treatment (1h, 100 $^{\circ}$ C) strongly impairs the enhancing activity of α -synuclein amyloid (new supplementary Fig. 1c). In contrast, EF-C fibrils remained active, which agrees with previous data showing that some amyloid fibrils are resistant to temperatures up to 300 $^{\circ}$ C (e.g. Surmacz-Chwedoruk et al., 2014; Ref. 50). Notably, monomeric forms of A β 40 and α -synuclein were used in all experiments as negative controls and lacked infection-enhancing activity (examples shown in the revised Fig. 6a and new supplementary Figs. 1b and 6).

5. The authors should define the abbreviation EF-C and clarify the rationale for its use. Is it a positive control?

The abbreviation and rationale for using EF-C are clarified in the revised manuscript (e.g. lines 60-62; 77-81).

6. Elaborate on the role of A β and α -synuclein in HIV infection to set up a premise for the study.

We expanded the introduction section to address this point.

7. The introduction effectively outlines the involvement of amyloids in various diseases but could benefit from a more detailed discussion on their dual roles in normal physiology and disease for a balanced view.

We agree that this dual role is important and mention it in the revised introduction (line 35/36 and lines 55-58). Since it's not the focus of this study and our manuscript already exceeds the usual length limit in-depth discussion of physiological roles of amyloids was not included.

8. How do the amyloid interactions described in this study compare with other known interactions in similar or different diseases?

To address this point, we tested and briefly discuss the potential impact of amyloids on Herpes Simplex virus 2 (HSV-2) and Zika virus that have been previously associated with neurological disease (new Figs. 5b-f; lines 216-229) and modified the discussion section (lines 326-328).

9. How specific are the interactions between α -synuclein or A β 40 fibrils and HIV-1? Could these interactions also affect other viral pathogens similarly?

These are interesting questions. To address them, we tested the effects of A β , α -synuclein, and EF-C fibrils on HSV-2 and Zika virus (new Figs. 5b-f). We found that treatment with increasing amounts of fibrils had little if any effect (new Figs. 5b, 5c). While all three types of amyloids allowed efficient precipitation of HIV-1, suggesting strong binding, ZIKV remained in the supernatant and HSV-2 showed an intermediate phenotype (Figs. 5d-f). Thus, our data indicate that the impact of the fibrils on HIV-1 is to some extent specific. As now discussed in the revised manuscript (lines 321-328), the accessibility of the viral membrane and intrinsic ability of viral particles for efficient attachment to target cells most likely play key roles. It is known that HIV-1 particles contain just a very limited number (i.e. ~7-14) Env trimers; thus, the viral membrane is readily accessible and our results indicate that amyloid fibrils impact HIV-1 infection more severely than other viral pathogens.

10. Authors need to discuss What molecular mechanisms underlie the increased HIV-1 infection rates in the presence of amyloid fibrils? What concentrations of amyloid fibrils were used, and how do these compare to physiological levels in neurodegenerative diseases?

We expanded the discussion section to address these points. In brief, we now clarify that amyloid fibrils promote viral attachment and fusion and why these effects are particularly strong in the case of HIV-1 (lines 304-328). We also discuss how the concentrations used compare to those detected in brain samples (lines 372-392).

11. Did the authors perform a dose-response study for each type of fibril to better understand the relationship between fibril concentration and HIV-1 infection enhancement?

The great majority of experiments (i.e. the data shown in Figs. 2a-d, 3a-c, 5b-f, 6e, 7, 8b and supplementary Figs. 1a, 1c, 1d, 1f, 2, 3, 6, 7) show dose response curves.

12. Please clarify the selection criteria for the cell lines and HIV strains used in the experiments.

We now provide more detailed information on these selection criteria for all viruses (lines 92-97) and cell types (e.g. lines 101-105, 115-117) used.

13. What are the potential therapeutic implications of the findings for HIV treatment or prevention, particularly regarding neurocognitive impairments?

We address this important aspect now in the last paragraph of the discussion section (lines 410-423).

14. Mention the key limitations of the study and how they might affect the interpretation of the results.

Limitations of the study are now specified in more detail (e.g. lines 369-373, 397-409).

15. The authors should discuss future studies to explore further the connections between amyloid fibrils and HIV infection or other neurological conditions.

We now provide a brief outline in the last paragraph of the discussion section (lines 410-423).

16. Minor grammatical errors need correction, and some sentences should be simplified for clarity and impact.

The text has been carefully read and edited by a native English speaker.

17. Detailed patient demographics should be provided better to understand the effects of these proteins on HIV replication.

Details of the demographics (age, sex distribution) of the patients are included in supplementary Table 1. Additional information (residency and post-mortem interval) was added in the revised version.

18. Clarify if the detrimental effects of A β and α -synuclein on HIV infection correlate with the severity of AD and PD.

As mentioned in the manuscript AD and PD occur substantially earlier and more frequently in HIV-1 infected individuals. Previous data suggest that HIV increases formation of A β and α -synuclein amyloids and we now discuss this in more detail (e.g. lines 384-392; 413-419). However, fully addressing this key question will require further studies.

Reviewer #2: The manuscript by Olari et al investigate an intriguing question of whether amyloid fibrils associated with Alzheimer and Parkinson disease, amyloid beta and alpha synuclein, enhance HIV infection in T-cells and myeloid lineage cells. Previous evidence the Semen-derived Enhancers of Viral Infection (SEVI), which promote HIV-1 infection, form amyloid fibrils and suggest that amyloid beta and alpha synuclein amyloid fibrils may act similarly in the brain and contribute to the enhanced neuropathogenesis of HIV. The premise of this manuscript is relevant to understanding the impact of aging in people living with HIV(PWH). This is increasingly important as the proportion of PWH in the US over the age of 50 continues to increase and will be the majority of the population in the coming decade. The manuscript includes promising data that there is enhanced activation of the HIV LTR in response to amyloid exposure. However, additional controls and assays are needed to draw the conclusions stated in this manuscript that alpha synuclein fibrils enhance HIV infection. The need for these controls is further supported by the data using patient samples that does not show specificity for alpha synuclein amyloid fibrils in induction of the HIV LTR. The following critiques are offered to help substantiate the conclusions drawn in the manuscript and support this novel hypothesis.

We are pleased that the reviewer appreciates the significance of our study. As specified below, some skepticism may have resulted from misunderstandings about the assay system used. Most of our studies show the impact of amyloid fibrils on infectious virus production by various cell types; however, we also examined attachment and fusion of viral particles. To improve clarity, we added schematics and additional textual information.

Major Critiques.

1. A key missing control for several experiments is the effect of the amyloid fibrils (AB, alpha synuclein and EF-C) without HIV on the viral reporter cell lines and primary cells. As the main assay for “HIV infection” is the readout of HIV LTR transcription via a reporter gene, it is important to demonstrate that the amyloid fibrils themselves are not activating the HIV LTRs in the reporter cells as the LTRs contain a number of cis-binding elements that are activated by NF-kB and early response elements that can be activated by numerous signaling cascades. There is a robust literature to suggest

that both beta amyloid and alpha synuclein interact with the toll like receptor and NLRP3 inflammasome pathways which would result in activation of the HIV LTRs even in the absence of viral infection. Without this control, it is difficult to specifically interpret the data from Figure 2 and the subsequent figures to be specific to HIV infection.

We had performed this control and agree that it is useful to include it. Thus, we now show that fibrils alone do not induce β -gal in TZM-bl reporter cells (revised Fig. 2a; supplementary Fig. 1a) and also included the “fibril only” control in the experiments with primary cells (revised Figs. 3b, 3c). Our results demonstrate that effects are due to HIV-1 infection and not the fibrils themselves.

2. It would be valuable to show the baseline effect of HIV infection alone for all assays to assess the robustness of the infection prior to the addition of the fibrils. The data are often presented as a fold change from baseline that precludes this assessment.

To address this point, we now provide the primary infection values for all infections experiments (Fig. 2c, 2d, 3a-c, 4c, 5b-f, 8; supplementary Figs. 1a, 1f, 2, 6, 7).

3. When using primary cells to infect with HIV, a critical complementary assay for HIV infection is production of p24. This assay would be important not only for Figure 3 to demonstrate that the primary cells are producing virus, but also for Figure 2C co-culture experiments to demonstrate that virus is produced from the infected cells prior to co-culture with the fibrils and the second cell type. Otherwise, the observation may be attributed to the activation or the inflammasome in the infected cell type and chemokine signaling activating the LTR in the second cell type. Controls to address this possibility should be added.

We agree that p24 is also a useful assay for HIV-1 production and have used both in previous studies. Our previous data showed that the results of both assays correlate (see e.g. Bosso et al., Fig. 3G; doi: [10.1016/j.celrep.2021.109735](https://doi.org/10.1016/j.celrep.2021.109735)). To further address this point, we now show that the enhancing effects determined by TZM-bl infection of p24 ELISA correlate very well (new supplementary Figs. 1c-e). As mentioned in the revised manuscript (lines 108-111), we used the well-established TZM-bl infection assay as it is more sensitive than the p24 antigen ELISA. In addition, it is linear over a broad range and infectious virus production the most relevant readout. For control, we now show that only background levels of β -gal activity are detectable in uninfected primary cells treated with the fibrils and in TZM-bl treated with the corresponding supernatants (revised Figs. 3b, 3c).

To clarify our assay systems, we added schematics in Figs. 2, 3, 4, 5 and 7. In addition, we clarified that HMC-3 cells are not productively infected in the coculture experiment but transfer the virus to the second HIV-1 permissive cell line (lines 128-145). Importantly, the readout in Fig. 2d is not LTR activation in TZM-bl cells, but a full-length infectious HIV-1 NL4-3 IRES-GLuc reporter construct that expresses the secreted Gaussia Luciferase. Altogether, inflammation and inflammasome activation play key roles in neurological disease but were not responsible for the induction of reporter gene activities and the enhancing effects of A β and α -synuclein on HIV-1 infection and replication.

4. Figure 3c suggests that the change in HIV LTR activation in microglia by co-incubation with amyloid fibrils is not related to HIV infection and exhibits the opposite kinetics to support this conclusion. The additional controls suggested may clarify this outcome and the interpretations and title of the manuscript should be modified to reflect that.

We respectfully disagree. In Fig. 3c, primary PBMC-derived microglia cells were infected with wild-type HIV-1 strains. As shown in the revised figure, production of infectious HIV-1 by primary T cells, macrophages and microglia determined by taking the culture supernatants at different time points and adding them to TZM-bl reporter cells. Neither supernatants from uninfected, fibril-treated

primary cells nor treatment of the various cell-types used with fibrils triggered significant reporter activity in the experimental settings used (Figs. 2a, 3b, 3c). As clarified in the revised manuscript (lines 176-186), our results are consistent with initial binding as well as productive infection of microglia. Early declining levels of virions released from the cells are measured followed by increases at later time point strongly indicate de novo HIV-1 production.

5. Use of HIV antiretroviral drugs are important controls that would be valuable in determining the contribution of HIV replication to the readouts (with appropriate controls added for the impact of these drugs on the cell types being used.)

To address this concern, we treated TZM-bl cells with Maraviroc or AMD3100 prior to infection with HIV-1 treated or not with fibrils. Maraviroc and AMD3100 efficiently inhibited R5 and X4 HIV-1 infection, respectively, even in the presence of A β , α -synuclein, and EF-C fibrils (new supplementary Fig. 2, lines 113-115). AMD3100 and Maraviroc did not affect the reporter in absence of the virus.

6. Caution should be exercised with the interpretations of the assay that only readout activation of the HIV LTR. As an example, in figure 4, the conclusion is drawn in lines 185-187 “These results support that amyloid alpha synuclein enhance HIV-1 replication in macrophages and microglia cells by enhancing virion attachment and fusion with these target cells.” as this is not demonstrated by simply observing the activation of the LTR in macrophage/microglia by expression of the reporter gene. Figure 5 tries to get at this; however, the use of murine leukemia virus to show interaction between the amyloid fibrils and this virus cannot be transferred to indicate that HIV interacts with the fibrils. The experiment would need to include a condition with HIV colocalizing with the amyloid on the surface of the infected cells.

We assume that the skepticism was most likely driven by a misunderstanding of the assay system. In Figure 4, we infected primary PBMC-derived macrophage and microglia cells with the macrophage-tropic HIV-1 AD8 strain carrying the Blam-Vpr reporter. As previously reported (Cavrois et al., 2002), the Blam-Vpr assay allows to directly determine HIV-1 fusion with target into the cells (see the explanatory schematic in the new Fig. 4a). Thus, it does not involve measurements of LTR activity. To address the second concern, we moved the MLV data to the supplement (supplementary Fig. 5) and performed similar experiments using CFP-labeled HIV-1 particles carrying the Env glycoprotein of the macrophage-tropic YU-2 strain. As now shown in the revised Fig. 5a, the amyloid fibrils colocalized with HIV-1 particles at the cell surface.

7. Due to the indirect nature of the assay in Figure 5 C, the conclusion drawn in lines 206-208, “Altogether, these results show that α -synuclein fibrils bind HIV-1 and promote virus infection similarly to EF-C amyloids, while A β fibrils are less effective.” are not substantiated by the data which support MLV interaction with the fibrils (5a/b) and show LTR activation kinetics of the fibrils mixed with HIV and exposed to reporter cells without the HIV on its own. The use of an HIV antiretroviral from the entry inhibitor class would aid in this conclusion.

As mentioned above, the TZM-bl assay is a well-established assay for quantification of HIV-1 infection and we excluded confounding effects of fibrils or other factors. Conclusions are now further confirmed by confocal microscopy of fluorescent HIV-1 particles (new Fig. 5a) and fusion assays (Fig. 4). In addition, we now show that binding to fibrils allows efficient concentration of infectious HIV-1 but not of Zika virus by low-speed centrifugation (revised Figs. 5b-f) and demonstrate that entry inhibitors prevent infection (new supplementary Fig. 2).

8. For several of the figures, it is unclear how the authors are determining a significant outcome as statistical methods are not outlined in the figure legends and significant findings are not indicated in the figure, particularly in Figures 2, 3, 5, and 6).

We added statistical information in the figures and their legends (also see reply to reviewer 1, point 1).

9. For Figure 7, important controls to show contribution of alpha synuclein would include cells that do not express alpha synuclein or the use of an alpha-synuclein neutralizing antibody controls to show alpha-synuclein specificity.

Such a control was included; i. e. the “ α -syn depleted” sample in Fig. 7 refers to cell culture supernatant where α -syn was depleted using beads coated with an α -syn antibody (lines 265/6). Additionally, we tested the supernatant of H4 cells transfected with an empty control vector. The results show that the enhancing effects on HIV-1 are α -synuclein specific (revised Fig. 7).

10. The human data do not support the hypothesis of the paper. The data are interpreted as not showing a difference between the extracts from patients with AD, LBD or control. However, as these data do not support the data using laboratory derived fibrils, this should be acknowledged and discussed instead of spun in a way to try to validate the findings from the laboratory derived treatments over the human derived materials.

As stated in the revised manuscript (lines 397-409) the results using human brain samples are very preliminary but may stimulate future studies. Samples numbers and quantities were limited and we could not determine which types of amyloids were present. In addition, the tissues originated from the occipital and temporal cortex of AD and LBD patients, which is not a hot spot of $A\beta$ and α -synuclein aggregation in diseased conditions. However, our results show that enhancement of HIV-1 infection correlates with the levels Thioflavin-stainable material in brain-derived samples (Fig. 8). This suggests that brain-derived amyloids are capable of enhancing HIV-1 infection, which agrees with our in vitro findings on synthetic fibrils. Definitive conclusions about the roles of different types of amyloids in HAND will require large human studies and are beyond the scope of this manuscript.

11. For Figure 8, controls to rule out the effects of non amyloid or non viral effects of human tissue on activation of the LTR are needed given the abundance of materials that could activate the NLRP inflammasome in the human lysates.

We thank this reviewer for this helpful comment. To address it, we incubated homogenized human brain tissue from four individuals without documented neurological complications with TZM-bl cells and confirmed that no reporter activity is induced in the absence of HIV (new Fig. 8d). Since no original material was left, we generated new extracts for this control.

12. The discussion should be modified to align with more cautious interpretations given the indirect nature of the assays interpreted as HIV infection and the use of transformed cell lines used for these studies.

As outlined above, the analyses shown in Figures 3 and 4 were done using primary T cells, macrophages and microglia cells. In addition to TZM-bl infection assays, we present direct measurements of virus attachment (Fig. 5), fusion (Fig. 4) and infectious virus production (Figs. 2, 3). We modified the discussion section and try to present all conclusions with great caution.

Minor points

Line 145-147 “These results suggest that amyloids may be more important for efficient spread of primary R5-tropic HIV-1 strains than for late-stage X4-tropic HIV-1 variants” This statement is an overinterpretation of the data that do not verify viral production. Further this statement should be for the discussion.

We omitted this statement.

Line 179-180 “Transfer into the target cells upon fusion is monitored by enzymatic cleavage of CCF2 by β -lactamase” It would be helpful to explain the biologic implication of this assay as opposed to the specific substrate/enzyme. This would aid in interpretation of the data for the reader.

We now provide a schematic of the assay (new Fig. 4a) and modified the text to clarify the biological implication (lines 191-194; 635-640).

Line 228 and 232, monogenic and heterogenic are not the best word as these are not genes, but protein mixtures.

We modified the text accordingly and changed the wording to “homogeneous” and “heterogeneous” (now lines 248 and 252).

Reviewer #3: In this study, Olari et al. report that incubating fibrils derived from alpha-synuclein or to a lesser extent Ab40 enhance HIV-1 infection. While the findings are interesting, they are not novel, the findings are undeveloped, there is a lack of detail, and many important controls are missing to support the conclusions being drawn. In addition, the underlying mechanism and biological relevance of the findings is unclear.

We are pleased that this reviewer feels that our findings are interesting and appreciate constructive criticism. However, many issues raised by this reviewer are difficult to comprehend and thus to address. Careful literature searches revealed that we overlooked one previous paper reporting that A β fibrils promote infection of TZM-bl and the leukemic T-cell line Molt-4 with derivatives of the lab-adapted NL4-3 strain (Widera et al. AIDS Research and Therapy 2014). We apologize for this oversight and now cite this study (Ref. 37; lines 62/63). To the best of our knowledge no studies on the impact of α -syn fibrils on HIV-1 infection have been reported and effects in primary human cells, including microglia remained to be determined. It is unfortunate that this reviewer did not provide specific information to substantiate his/her concerns and misunderstood or missed some data.

Main Concerns

It is well established that fibrils enhance infection and some studies of HIV and alpha-synuclein are not cited. The vast majority of this study just involves mixing virus preparations with fibrils and simply showing the final effect on infectivity, but the preparations and relevance are not characterized at all. From the methods, it seems that the virus and fibril are just mixed and added to cells. The mixed preparations need to be characterized in detail. How much virus is bound to fibrils and how much is not? How much fibril is not bound to virus?

We and others have established that semen-derived amyloids enhance HIV-1 infection. As mentioned above, we missed one previous study on the effect of A β and now mention and cite it (Ref. 37). However, this is the first study to show that α -synuclein fibrils enhance HIV-1 infection. Information on the morphology and surface charge of the fibrils is provided in Fig. 1. Essentially all results

presented assess the relevance of the fibrils, i.e. effects on HIV-1 attachment, fusion, infection and replication. Pull-down experiments show that essentially all infectious HIV-1 is bound to the fibrils in contrast to Zika virus and HSV-2 particles (revised Fig. 5). In the revised manuscript, we performed confocal microscopy analyses showing that complexes between HIV-1 virions and amyloids are detectable at the cell surface (new Fig. 5a).

The images in Figure 5 do not address this issue and instead raise more questions. First off, they switch to MLV rather than HIV. Second, the images are poor quality as, for example, the cytoskeleton stain is completely lacking in detail and just shows a huge blob where the cell is. The individual images are not from the same fields of view in, for example, the top panels in 5a, or if they are, it is not clear that they are and whether the MLV fibril panel is a zoom of a specific region. There is no scale bar on this particular image, but from this and other images, the fibrils relative to the size of virus particles or cells seem enormous compared to the sizes shown in Figure 1. This would suggest that the fibrils are being added in enormous excess. Is this physiologically relevant? How do they promote infection? Do they activate cell signalling, for example? The proposed mechanism is an assumption and based on prior literature, making it uncertain and not particularly novel.

To address this issue, we performed experiments using CFP-labeled HIV-1 particles which confirmed their fibril- and cell-binding activity (new Fig. 5a). The MLV images were moved to the supplement (Fig. S5) and scale bars shown in all images. In the new microscopic analysis, we stained the nuclei with CellTrace and the actin skeleton with ATTO-phalloidin. Confocal microscopy will only detect larger aggregates but the average fibril size is provided in Fig. 1e. High concentrations of fibrils in the brain are a characteristic of neurodegenerative diseases. The proposed mechanism is further supported by data on effects on HIV-1 binding, virion fusion, infection and virus replication as well as previously literature showing that HIV-1 infection is enhanced by the particles binding to fibrils, and not due to cellular signaling. Quantities and potential relevance are discussed in the revised manuscript (lines 372-392).

During a natural infection, how much alpha-synuclein is made and is it secreted? How is it made during infection? Or is this proposed to be a passive consequence of infection of the brain where these products are made? In which case, what are the physiologically relevant levels of these fibrils as they are unlikely to be as abundant as shown in figure 5.

This reviewer may have misunderstood. We are not proposing that HIV-1 infection induces the production or secretion of alpha-synuclein, but that the fibrils enhance the infection. We now discuss secretion and the levels of α -synuclein detected in vivo, as well as effects of HIV-1 infection in the revised discussion. Notably, α -synuclein is highly abundant and the relevance of cell-free α -synuclein increasingly recognized. Determination of the physiological levels of α -synuclein in patients is a challenging task because of the limited availability of brain material and since local rather than overall concentrations are relevant for detrimental effects of amyloids.

The studies of EF-C and how it enhances fibril formation have little relevance to actual infection as it is not clear that EF-C is made naturally. My understanding is that it is an artificial peptide used to promote retroviral transduction.

As mentioned in the original and revised manuscript (e.g. lines 58-62; 77-81). EF-C is a fragment of the HIV-1 Env gp120 protein and related amyloidogenic Env fragments have recently been detected in the CSF of AIDS patients. While the in vivo relevance remains to be determined, Env-derived amyloids may contribute to HAND by both promoting HIV infection and cross-seeding other amyloids (Fig. 6, lines 297-299; 352-368).

The use of gaussian luciferase-based fibril transfection systems is interesting but artificial and similar to the mixing experiments, is limited in terms of scope, characterization and relevance.

The reviewer may not be familiar with our assay systems. In this assay, cells are transfected with plasmids encoding the monomeric form of α -synuclein fused to a split Gaussia luciferase. Upon oligomerization of α -synuclein the split Gaussia is becoming active. The results show that cell-produced α -synuclein significantly enhances HIV-1 infection, similarly to externally added-synthesized α -synuclein. As this is a broadly used, state-of-the-art in vitro method and the general negative comments of this reviewer are not substantiated.

The trans-infection assays need to be better controlled. The authors state that the microglia cell line that is used cannot be infected, but monocyte-derived microglia (MMGs) can. But there could be a low level of infection in line used for the trans-infection assay. This assay extends for a much longer period of time and although it uses TZM-bl as recipient cells, the assay uses viral gaussian as a readout of infection, which does not tell you which cell type is infected. Throughout the paper, there is no indication of MOI or the titers of different virus stocks and strains. If the microglia line is just infected at less efficiently than other cell lines, the virus may be spreading in them slowly and infecting TZM cells at the same time during this longer experimental setup. There is no evidence for the claim of trans-infection in this setup, or of the general conclusions being made.

The reviewer is correct that HIV-dependent Gaussia expression in the co-culture experiment does not allow to determined which cell type is infected. However, s/he may have missed that we performed the same assay with only HMC-3 cells for comparison and control (Fig. 2c). Although this assay is highly sensitive, no increase in the gaussian signal was observed, while it increased by up to three orders of magnitude in the presence of TZM-bl cells (Fig. 2d) providing clear evidence for trans infection of the latter. As mentioned in our manuscript (lines 128-145), these results agree with the previous findings that HMC-3 cells are not susceptible to HIV infection and that semen-derived fibrils promote in trans infection of HIV-1. As this reviewer may not be familiar with HIV-1, we feel there is need to state that MOIs and titers are cell-type dependent and hardly used in the HIV field, instead assays like TZM-bl infection or p24 antigen ELISA are considered state-of-the-art.

Antibody-mediated depletion is used in several approaches but in no case is the depletion efficiency demonstrated and control depletions or antibodies are missing. Similarly, where peptides or fibrils are added, control peptides rather than nothing at all should be added.

Antibody-mediated depletion was only used in one experiment and efficient depletion of α -synuclein was evident from the loss of HIV-1 enhancing activity (Fig. 7). To verify that effects are dependent on α -synuclein, we transfected the cells with an empty plasmid (revised Fig. 7). This reviewer missed or ignored that monomeric peptides were used as controls in all experiments.

The use of human brain homogenates is interesting but again, they are heterogenous in nature and poorly characterized and controlled. These preparations contain a lot of things that could influence infection. The depletion approach is again uncontrolled. There is no evidence of actual depletion or the extent to which depletion occurs to correlate with effects on infection. Control antibodies need to be used also. The effects are just correlative in their current state.

This part was a small pilot study providing suggestive evidence that the overall amount of amyloid in samples derived from human brains correlates with the enhancing effect on HIV-1 infection (Fig. 8). Larger future studies are required to distinguish between different types of amyloids and to unravel potential links with human diseases. Other than assumed by reviewer 3, no aggregate depletion was performed. In fact, amyloid aggregates most likely account for the effects on HIV-1 infection.

THT is a very broad fibril staining approach that does not discriminate between different types, particularly when using it to characterize these brain extracts. More specific measures of each fibril type are needed. In addition, where THT is used in experiments, does it affect fibril structure or binding to virus?

We explicitly state in the manuscript that we did not use THT staining to discriminate between different types of fibrils (lines 400-409). The quantities of human brain extracts were very limited and it is beyond the scope of this study to distinguish between different fibril types. Thioflavin T was used to assess the quantity of amyloid in the extracts and not present in experiments involving HIV-1; thus, a potential impact on virus binding is irrelevant.

Several experiments are performed just once and measured several times to generate data points and statistics. In other cases, the number of independent repeats is unclear. For imaging in figure 4, for example, 5 different images are analysed which is not a lot, and it is unclear if this experiment was done just once.

The great majority of experiments were independently performed three times, each in duplicate or triplicate and this is specified in the figure legends. As noted by the reviewer, the imaging experiment shown Fig. 4 was performed just once but five different images quantified and the results significant. Due to the limited availability of brain tissue, the experiment shown in Fig. 8b was only performed once, but in triplicates at four different doses to ensure robustness of the measurements.

Minor Comments

Brain-derived HIV strains may be better for several of the cell types being studied.

We provide a more in-depth rationale for the choice of HIV-1 strains in the revised manuscript (lines 92-97). Amongst others, we used the brain-derived JR-CSF infectious molecular clone.

The authors use soluble fibrils but, in many cases, fibrils are thought to be insoluble and form plaques. This should be discussed to better understand the relevance of the choice.

Plaques are the end result of a long process, which begins with the production and release of A β or α -synuclein monomers. These monomers aggregate into toxic oligomers, which then form protofibrils, fibrils, and ultimately, dense plaques characteristic e.g. of Alzheimer's disease. Our approach mimics the first steps of this process from monomers, over oligomers to amyloid fibrils.

Line 77: EF-C amino acids numbers in the text does not match that of the table in Figure 1a.

We apologize for this mistake and corrected it.

Line 88-82 is confusing, are the authors making a structural comparison between the fibrils?

We modified the text for clarity.

Suppl Figure 3b: The staining for P2RY12 is not strong and is unconvincing. It seems equivalent in both macrophages and microglia, which does not align with the claim that it is only expressed by microglia or what is shown for the FACS analysis.

As this reviewer may know, not all antibodies work equally well in microscopy and FACS analyses. The FACS data in suppl. Figure 4a are clear and provide a more quantitative readout than the microscopy data shown in suppl. Figure 4b.

Reply to the reviewer`s comments (in italics)

Reviewer #1 (Remarks to the Author): The authors revised the manuscript satisfactorily.

We thank reviewer 1 for helping us to improve our study and the positive feedback.

Reviewer #3 (Remarks to the Author): The authors approach of undermining the reviewer`s expertise and understanding is not particularly helpful (both this reviewer and others), and in some cases seems to be used to avoid addressing comments. Despite this, the revised manuscript now better explains the potential significance. However, I do think some issues remain that need to be clarified.

We are pleased that this reviewer now better understands what was done and why it is important. We hope that our replies below will clarify the remaining points.

The authors avoid addressing comments about the quality of images of MLV and fibrils in figure 5, and simply move it to the supplemental data. The regions at the same magnification (namely 10um bars) are from separate images and the merges are confusing. For example, in Supplemental Figure 5a the image of “MLV, Cells” and “MLV, fibrils, Cells” is clearly not a merge of the “MLV, fibrils” image, as the fibrils are different in each image, as is the MLV alone image. This is unconventional and confusing, what is being presented here? The same is occurring for supplemental 5b.

Reviewer 3 previously criticized that we analyzed MLV and not HIV and that we didn't stain the cytoskeleton. To address this, we analyzed HIV particles and stained the actin skeleton (new Figure 5a). To make space for these higher resolution images in the main paper, we moved the original MLV data to the supplement (Figure S6a). Our images are of high quality (600 dpi in the submitted PDF). The question addressed in these experiments was whether fibrils bind virions and promote their attachment to cell surface. Our results obtained with both MLV and HIV particles clearly confirm this. To illustrate this point, we show enlarged insets of virus/fibril complexes at the cell surface (Figures 5a, S6a). To avoid any confusion, we now show all channels separately, in the new supplemental Figures 5 and 6b.

The new images of HIV in figure 5b are just merges that lack detail. It looks like HIV forms huge clumps presumably on fibrils, but it is unclear how much of the orange signal is from the fibrils versus overlap with virus particles. It cannot be 100% overlap and separate images should be shown of what HIV particles look like under these conditions, as is done for MLV.

The resolution limit of light microscopy does not allow to discern individual viral particles/fibrils. Thus, CFP fluorescence emitted by particles that in close proximity to each other may appear as clumps. Like for MLV, the images do NOT just represent merges but analyses of HIV particles, cells and fibrils alone or in combination. As outlined above, we now also show data in a more conventional way and as separate channels in the new supplemental Figures 5 and 6b to completely rule out any possible misunderstandings.

The authors argue that they did some experiments once but analyzed five cells. If the experiment itself had an issue then all five cells are erroneous, as is the interpretation. The whole experiment should be repeated at least a second time to ensure this is not the case, or discuss this in light of other findings that support the conclusion independently.

In the fusion experiment, we did not analyze five cells but five different images, each representing hundreds of cells. To further address this point, we repeated the experiments completely independently twice and confirmed that the results are highly reproducible (see examples below). We specified in the legend to Figure 4 that results show representative data and were confirmed in two independent experiments (lines 1059-1060).

Correlation between the mean fluorescence intensities obtained in independent HIV-1 BlaM-Vpr fusion assays. Experimental details are provided in the legend to Figure 4.

The authors argue that the depletion approach for α -synuclein must have worked because it reduced HIV infection to avoid addressing the question of verifying that the depletion worked, was specific and correlates with the effect on infection. Reduced infection could be due to any number of issues, such as the antibody itself neutralizing the virus, and it seems important to verify that the extent of α -synuclein depletion correlates at least to some extent with the effects on infection.

As clarified in our previous reply to this reviewer, we provide two lines of evidence that the effects are specific for α -synuclein; i.e. neutralization by α -synuclein specific antibodies and by analysis of the supernatant of cells transfected with a control vector. Since conditions were otherwise identical, it is difficult to comprehend which other "number of issues" should affect infection. To satisfy this reviewer, we tested whether the antibody to α -synuclein neutralizes HIV and -as expected- confirmed that it is not the case.

The trans-infection assay remains quite crude and overinterpreted, as noted by other reviewers. The transfer event could very easily be from the virus bound to fibrils on the surface and nothing to do with the cell type transferring virus. Adding fibrils alone, without cells, will also increase infection so the HCM-3 cells could simply be fibril carriers. Overall, the interpretations are speculative particularly as the authors use a singular readout of infection for two different mixed cell types, and the biological relevance is unclear. This assay doesn't add much to the report.

The other reviewers had no remaining concerns about the trans-infection assay and reviewer 3 may have misunderstood the assay setting and interpretation. We agree that the "transfer event could very easily be from the virus bound to fibrils on the surface" and that infection is "simply" enhanced because HCM-3 cells are "carriers of fibril/virion complexes. We never propose that this effect is cell-type dependent. In fact, we have previously reported that semen-derived fibrils promote trans infection of HIV by other cell types (Münch et al., Cell 2007). Thus, it remains unclear to which "speculative interpretations" this reviewer is referring. To avoid any potential future misunderstandings, we mentioned in the revised manuscript (lines 349-357) that the physiological relevance of binding of virus-fibril aggregates to cells that are not susceptible to infection remains to be determined. However, transfer of the virus to susceptible cells might be high biological relevance, especially if the fibril/virion carriers are highly migratory, like microglia cells.

Reviewer #4 (Remarks to the Author): This is a highly interesting extension of previous findings (Wojtowicz et al. 2002) that amyloid-derived peptides enhance the attachment and infection by HIV-1 and other enveloped viruses including MuLV. Furthermore, Münch et al. (Dr. Kirchoff group) in a Cell paper (2007) previously showed that amyloid fibrils in semen enhanced HIV-1 infection. The current findings now extend this finding to show that α -synuclein, a presynaptic protein linked neuropathologically to Parkinson's disease, also enhances HIV-infection, possibly by the same mechanism of increasing adsorption or adherence to cell membranes, but not A β 40. The results are not entirely unexpected, but do point to the impact of amyloid fibrils in novel settings such as Parkinsons and HIV.

We thank reviewer 4 for the positive assessment and constructive suggestions that helped us to improve our manuscript.

In the introduction section, it is stated that “the levels of amyloid deposition correlate with viral loads” – with a citation of two review articles. This is a very important finding that would set the stage for the current manuscript. The authors should cite the original references. It was impossible for this reviewer to find the original references within the two review articles cited.

To address this, we further examined the primary literature. While several studies reported increased amyloid levels in HIV-infected individuals indeed none of them showed a definitive correlation between these levels and viral loads. Thus, we thank the reviewer for making us aware of our mistake, rephrased this part and mention additional original references reporting on the possible reasons for increased A β amyloid formation in HIV infection (lines 43-49).

The manuscript essentially shows that a-synuclein, but not Abeta40 fibrils (but not their monomers), enhance HIV-infection of a variety of target cells including CD4 T cells, monocyte derived macrophages and monocyte-derived microglia. The primary influence of a-synuclein fibrils appears to be in enhancing the virion attachment to the membrane. The binding or capture of HIV to the plasma membrane can be achieved even without CD4 receptor, but in that case it remains bound to the cell but does not enter the cells. But if CD4 and a coreceptor are present, the virus appears to proceed and complete the infection. The fluorescence images show an increased attachment of the virus, when a-synuclein fibrils are present, to the cell surface. Vpr-beta lactamase-based fusion assay results show that the fusion is enhanced. Although the fusion is the downstream step, this is expected.

I failed to see the importance of the findings reported in Figure 6. As a technique to enhance the infection by HIV-1, the role of EF-C can be appreciated (already published and cited in the manuscript). But the question being addressed here, whether EF-C enhances the rate of aggregation of either Amyloid or the a-synuclein – appears something out of a pure technical point of view. It does not add to the mechanistic understanding of the phenomenon being reported.

As now mentioned in the manuscript (lines 235-237) amyloidogenic HIV-1 Env fragments similar to EF-C have been detected in CSF of people living with HIV (ref 44) and it has been reported that infusion of gp120 induced accumulation of amyloid plaques in mice (ref 100). EF-C forms amyloids with high efficiency (ref 37) and HIV infection has been reported to accelerate the development of neurological disease. Thus, it seems a relevant question whether fragments of HIV-1 Env may promote the formation of A β and a-synuclein fibrils. We agree, however, that the underlying mechanisms and physiological frequency remain to be determined. To address this point, we now provide a clearer rationale for the cross-seeding experiments in the revised manuscript (lines 235-237) but also state their limitations (lines 373-380).

The responses of the authors to the Reviewer 2 comments are all acceptable – except those for #10. In their responses to point #10 by Reviewer 2 – “The human data do not support the hypothesis of the paper” and “this should be acknowledged and discussed instead of spun in a way to try to validate the findings from the laboratory derived treatments over the human derived materials” – the authors commented as follows. Author comments are in quotes followed by my remarks.

a. “the results using human brain samples are very preliminary”. b. In addition, they “could not determine which types of amyloids were present” c. “Definitive conclusions about the roles of different types of amyloids in HAND will require large human studies” d. “results show that enhancement of HIV-1 infection correlates with the levels Thioflavin-stainable material in brain-derived samples”

On the one hand the authors appear eager to present information that agrees with their in vitro data. On the other hand, they know that their human results are very preliminary and have many caveats. Therefore, I agree with the Reviewer 2 on this point. The authors should acknowledge these shortcomings (a, b, c and d above) in the text and allow the reader to draw their own conclusions.

We are pleased that reviewer 4 feels that we properly addressed all but one of the points raised by reviewer 2. We hope that our results obtained using primary brain extracts will stimulate future studies. However, we are aware that these findings are preliminary and now clearly mention the limitations in the revised manuscript (lines 291-292 and 305-307).